# Constructing functional models from biophysically-detailed neurons

**Peter Duggins***, **Chris Eliasmith**

Computational Neuroscience Research Group, Department of Systems Design Engineering, University of Waterloo, Waterloo, Canada

* psipeter@gmail.com

## Abstract

Improving biological plausibility and functional capacity are two important goals for brain models that connect low-level neural details to high-level behavioral phenomena. We develop a method called "oracle-supervised Neural Engineering Framework" (osNEF) to train biologically-detailed spiking neural networks that realize a variety of cognitively-relevant dynamical systems. Specifically, we train networks to perform computations that are commonly found in cognitive systems (communication, multiplication, harmonic oscillation, and gated working memory) using four distinct neuron models (leaky-integrate-and-fire neurons, Izhikevich neurons, 4-dimensional nonlinear point neurons, and 4-compartment, 6-ion-channel layer-V pyramidal cell reconstructions) connected with various synaptic models (current-based synapses, conductance-based synapses, and voltage-gated synapses). We show that osNEF networks exhibit the target dynamics by accounting for nonlinearities present within the neuron models: performance is comparable across all four systems and all four neuron models, with variance proportional to task and neuron model complexity. We also apply osNEF to build a model of working memory that performs a delayed response task using a combination of pyramidal cells and inhibitory interneurons connected with NMDA and GABA synapses. The baseline performance and forgetting rate of the model are consistent with animal data from delayed match-to-sample tasks (DMTST): we observe a baseline performance of 95% and exponential forgetting with time constant $\tau = 8.5s$, while a recent meta-analysis of DMTST performance across species observed baseline performances of $58 - 99\%$ and exponential forgetting with time constants of $\tau = 2.4 - 71s$. These results demonstrate that osNEF can train functional brain models using biologically-detailed components and open new avenues for investigating the relationship between biophysical mechanisms and functional capabilities.

## Author summary

Computational models of biologically realistic neural networks help scientists understand and recreate a wide variety of brain processes, responsible for everything from fish locomotion to human cognition. To be useful, these models must both recreate features of the brain, such as the electrical, chemical, and geometric properties of neurons, and perform

**Data Availability Statement:** All relevant data and code are available on GitHub (https://github.com/psipeter/functional-detailed-neurons).

**Funding:** This work was supported by the Canadian Foundation for Innovation (52479-10006,

CE), the Ontario Innovation Trust (35768, CE), the Natural Sciences and Engineering Research Council of Canada (261453, CE), and the Air Force Office of Scientific Research (FA9550-17-1-0026, CE). The funders had no role in study design, data collection and analysis, decision to publish, or preparation of the manuscript.

useful functional operations, such as storing and retrieving information from a short term memory. Here, we develop a new method for training networks built from biologically detailed components. We simulate networks that contain a variety of complex neurons and synapses, then show that our method successfully trains them to perform a variety of cognitive operations. Most notably, we train a working memory model that contains detailed reconstructions of cortical neurons, and demonstrate that it performs a memory task with performance that is comparable to simple animals. Researchers can use our method to train detailed brain models and investigate how biological features (or deficits thereof) relate to cognition, which may provide insights into the biological basis of mental disorders such as Parkinson's disease.

# 1 Introduction

Biological detail is an important concern for neural models that seek to bridge the gap between neural and cognitive processes. The inclusion of synaptic and cellular mechanisms in cognitive models allows researchers to investigate aspects of psychology, such as biologically-grounded mental disorders and their pharmacological treatment, that often cannot be studied with simpler models. Biological features, such as neural nonlinearities and local connectivity, have historically inspired innovations in artificial intelligence, including convolutional networks and long-short-term-memory units. Unfortunately, simulating biological detail often increases the difficulty of building and analyzing cognitive models. To mitigate these difficulties, most models compromise on either biological realism, such as ion-channel dynamics or synaptic connectivity, or cognitive capacity, defined as the ability to perform computationally-useful operations on inputs and produce task-relevant behavioral outputs. For example, cognitive architectures like ACT-R [1] have produced numerous models of human cognition that closely match behavioral data, but which use production rules and activation functions that coarsely approximate the brain's neural substrate, limiting their ability to investigate many neurocognitive phenomenon. At the opposite extreme, the Human Brain Project [2] has produced models that recreate the anatomy and spiking behavior of entire cortical microcircuits, but these models do not perform recognizable neural computations or produce behavior, limiting their utility for investigating cognitive phenomenon. Other models, such as SPAUN [3], use neural networks to realize cognitive systems and perform tasks end-to-end, but have limited biological realism: while they respect the anatomical connectivity between brain structures and recreate their proposed functions, many of their low-level mechanisms, such as the neuron and synapse models, are crude approximations of the brain's biological substrate. To study the biological foundations of the human brain and design biologically-inspired cognitive algorithms, models which unify biophysical detail and cognitive capacity are needed.

Several existing frameworks leverage biological neural networks to perform cognitive tasks. The Neural Engineering Framework (NEF) and the Semantic Pointer Architecture [4, 5] use large-scale, biologically-constrained spiking neural networks to study the functional aspects of cognition. Similarly, the Leabra cognitive architecture [6] uses biologically-plausible learning rules to capture the functionality of major cognitive systems. Other approaches emphasize dynamics within neural networks, which, when properly controlled, can be used to perform mathematical transformations of represented information and issue behavioral commands. The efficient coding hypothesis [7] uses cortical connectivity to produce controlled spiking networks, while FORCE [8] uses online learning rules in recurrent networks to implement dynamical systems. While each of these frameworks emphasizes the role of particular

biological features (such as spiking neurons, learning rules, or constrained connectivity) for cognition, little attention has been paid to the complexity of individual neurons, the fundamental unit in most intelligent systems. For the most part, these architectures assume simple models for neurons, ranging from rate-mode neural assemblies to point neurons like the leaky-integrate-and-fire model.

This raises the question: how important are the biological details of individual cells for cognitive systems? It is certainly true that that some cellular features, such as cellular respiration, have little bearing on information processing, while others, like dendritic structure and calcium dynamics, play a larger role. Computational models are an essential tool when investigating the contribution of these cellular mechanisms: they allow researchers to systematically vary the network's biological complexity while examining changes in cognitive performance. At a minimum, experimentation with such models would show that specific biological features do not affect a network's functional capabilities: this result would justify the exclusion of these features for future research, validating the current paradigm of simple, easy-to-simulate neuron models. On the other hand, experimentation may reveal that existing architectures do not function properly when cellular complexity is reintroduced, which would suggest that theoretical improvements are needed to explain how brains accommodate this complexity. Modellers may even find that adding biological complexity increases the computational power or cognitive flexibility of the network, for example through dendritic filtering or neurotransmitter-dependent spike modulation.

In this paper, we propose a method for training and simulating networks that contain biologically detailed neuron models and perform useful cognitive operations. We label this method "oracle-supervised Neural Engineering Framework" (osNEF): it is an extension of core NEF principles that utilizes an "oracle", a parallel network that is used during training to supervise the learning process. One major advantage of osNEF is that it treats the neuron model as a "black box", relying on learning rules that only consider the spiking inputs and outputs to each cell. Because it does not rely on detailed knowledge of cellular dynamics, osNEF can be applied to a wide range of neuron models without major changes to the algorithm. To facilitate easy adoption of osNEF, we develop an interface where modellers may plug in existing neuron models, written in Python or NEURON, to the general-purpose Nengo simulator [9], which facilitates large-scale functional modelling. This approach lets modellers specify detailed low-level mechanisms like conductance-based synapses, voltage-gated ion channels, and dendritic geometry, then train the network to realize high-level dynamics or computations. Our goal is to show that osNEF can be used to construct a variety of functional neural networks from various biologically detailed components, allowing researchers to investigate questions that relate low-level biological details (e.g. calcium dynamics) to high-level cognitive capacity (e.g., task performance).

To demonstrate these capabilities, we apply osNEF to produce two classes of cognitive functionality. First, we highlight the broad applicability of osNEF by simulating networks of biologically detailed neurons and training them to implement specific dynamics. We assume that some low-dimensional state space is represented by heterogeneous spiking activity within the network, then show that synaptic transmission between groups of detailed neurons may realize specific mathematical operations on that state space, notably addition, multiplication, oscillation, and integration. These operations were chosen because they represent computational primitives that are widely used in cognitive systems. We simulate four different neuron models, ranging in complexity from a LIF point neuron to a spatially-extended pyramidal cell, and show that osNEF can accommodate them all to realize the target operations. Second, to demonstrate a concrete cognitive application, we construct a biologically-detailed model of working memory in PFC that performs an idealized memory task. The network is composed of

anatomically-detailed reconstructions of layer-V pyramidal cells and fast-spiking interneurons; both neuron models contain numerous geometric compartments and ionic currents and are connected using conductance-based NMDA and GABA synapses. We show that the mnemonic performance of the model is consistent with empirical data from a standard test of working memory, the delayed match-to-sample task (DMTST): both simulated and empirical data are well-characterized by an exponential forgetting curve ($y(t) = B \exp(-t/\tau)$) with baseline performance $B = 80 - 100\%$ and forgetting time constant $\tau = 10 - 60s$. We conclude with a discussion of the strengths and limitations of osNEF, including its biological plausibility and cognitive generality, and by comparing it to similar approaches.

## 2 Background

As in the NEF and SPA, we characterize spiking activity within populations of neurons as encoding information in a latent *state space*. While spikes are the physical means of communication between neurons, cognition can be analyzed (to a large degree) as transformations of these lower-dimensional states, permitting a more abstract, computational, or symbol-like description of what brains do. We assume that this state space can be represented by a vector-valued signal $\mathbf{x}(t)$, and that the cognitive operations performed in the brain may be described through the dynamics of this state space $\dot{\mathbf{x}}(t)$. At the sensory periphery, neurons transduce external signals (light, sound, etc.) into spikes which represent the stimuli. This representation is somehow transformed via connections within the brain, ultimately producing motor commands that activate the body's muscles. Because we are chiefly concerned with cognitive processes, we will also treat sensory inputs and motor commands as state space signals, and focus primarily on how spiking activity and neural connections represent and transform $\mathbf{x}(t)$ within the brain.

Broadly speaking, our goal is to describe cognitive processing in terms of state space dynamics, then train the connectivity within a neural network such that its spiking activity implements those dynamics. As in the NEF, we first define encoding and decoding between neural activity and the state space, then later describe how synaptic connections implement state space transformations. Given a signal $\mathbf{x}(t)$ and a population of neurons, the signal must drive those neurons to fire in patterns that represent the signal. Each neuron spikes most frequently when presented with its particular "preferred stimulus" and responds less strongly to increasingly dissimilar stimuli (i.e., values of $\mathbf{x}(t)$). Each neuron $i$ is accordingly assigned a preferred direction vector, or *encoder*, $\mathbf{e}_i$. At the sensory periphery, $\mathbf{e}_i$ determines how the external $\mathbf{x}(t)$ is transduced into electrical inputs to neuron $i$; within the network itself, $\mathbf{e}_i$ co-determines connection weights, which dictate how neuron $i$ responds to spiking inputs from an upstream population whose activities represent $\mathbf{x}(t)$. Mathematically, both these processes can be summarized as

$$J_i(t) = \mathbf{e}_i \cdot \mathbf{x}(t), \tag{1}$$

where $J_i(t)$ is the electrical current driving neuron $i$, and $\mathbf{e}_i \cdot \mathbf{x}(t)$ is the dot product between the state space input and neuron $i$'s encoder. This current drives the neuron model,

$$a_i(t) = G[J_i(t) + \beta_i], \tag{2}$$

where $a_i(t)$ is the spiking activity, $G$ is the neuron model, and $\beta_i$ is an optional bias current. The specifics of how inputs drive the cell, and how the cell dynamically responds, vary across neuron models $G$, as discussed below. A distributed encoding extends this notion: if $\mathbf{x}(t)$ is fed into multiple neurons, each with a unique tuning curve defined by $\mathbf{e}_i$ and other parameters,

then each neuron will respond with a unique spiking pattern $a_i(t)$, and the collection of all neural activities will robustly encode the signal.

For state space representation to be useful, there must be methods to recover, or decode, the original vector from the neuron activities. We identify *decoders* $\mathbf{d}_i$ that either perform this recovery or compute arbitrary functions, $f(\mathbf{x})$, of the represented vector. A functional decoding with $\mathbf{d}_i^f$ allows networks of neurons to *transform* the signal into a new state, which is essential for performing cognitive operations. To compute these transformations, a linear decoding is applied to the neural activities:

$$\hat{f}(\mathbf{x}(t)) = \sum_{i=0}^{n} a_i(t) \; \mathbf{d}_i^f, \tag{3}$$

where $a_i$ is the activity of neuron $i$, $n$ is the number of neurons, and the hat notation indicates that the computed quantity is an estimate of the target function. Neural activity $a$ is obtained by convolving spike trains with a smoothing filter $h$:

$$a_i(t) = \sum_{T} h(t - T) * \delta_i(T) \tag{4}$$

where $T$ are the spike times, $h$ is an impulse response function for the filter, ($*$) is convolution, and $\delta_i$ is a Dirac delta function describing the spike train for neuron $i$. To maintain correspondence with biology, we use these same filters $h$ when implementing synapses within the network (see Sec 3.3); we therefore typically refer to $h$ as the "synaptic" filter.

At the motor periphery, $\mathbf{x}(t)$ is retrieved using appropriate decoders, driving the model's behavioral output; within the network itself, $\mathbf{d}_i^f$ co-determines connection weights, which dictate how downstream neurons respond to spikes produced by neuron $i$. Connection *weights* between each presynaptic neuron $i$ and each postsynaptic neuron $j$ combine encoders and decoders into a single value: the weight matrix $\mathbf{w}$ is the cross product between the encoder and decoder matrices,

$$\mathbf{w} = \mathbf{e} \times \mathbf{d}^f. \tag{5}$$

Our primary tool for studying state space transformations is control theory, which specifies the dynamics of $\mathbf{x}(t)$ as

$$\dot{\mathbf{x}}(t) = A\mathbf{x}(t) + B\mathbf{u}(t), \tag{6}$$

where $\dot{\mathbf{x}}(t)$ is the derivative of the represented state, $\mathbf{u}(t)$ are inputs (external signals or upstream representations), and $A$ and $B$ are the transformation matrices. $B$ governs how feedforward inputs affect the current representation (e.g., scaled addition), while $A$ governs how recurrent inputs affect the representation (e.g., simple harmonic oscillation); both $A$ and $B$ are implemented through weighted synaptic connections between neurons in populations representing $\mathbf{u}(t)$ and $\mathbf{x}(t)$. In this work we restrict ourselves to linear dynamical systems; while the theory and mathematical assumptions of the NEF and osNEF do not confine these frameworks to linear systems, demonstrating that osNEF can successfully realize the wider (and more difficult) class of nonlinear systems is a significant extra challenge that we leave for future work.

Given this framework, which is a direct recapitulation of the NEF [4], the challenge of building cognitive neural systems reduces to (a) describing a cognitive algorithm as a dynamical system (e.g., Eq 6), and (b) finding encoders and decoders such that neural connection weights implement the transformations dictated by the dynamical system. In addition, learning rules based on error-driven feedback or supervision may be used to train the weights

through encoder and/or decoder updates, in which case the resulting cognitive algorithm may be estimated once learning is complete. The framework is therefore consistent with either a top-down or bottom-up approach to neural modelling, and our method combines both approaches when learning the parameters of synaptic connections.

## 3 Methods

We now summarize the osNEF, which extends the core NEF by introducing new methods for training neural networks, and describe the neuron models we tested; all code is available on GitHub.

### 3.1 Tuning curves

We begin by describing a target *tuning curve* which relates a neuron's input $\mathbf{x}$ to its spiking activity $a$. This tuning curve should include a nonlinearity so that networks of neurons with such tuning may compute interesting functions on state space inputs. Many target tuning curves are possible: we seek one that is both easily parameterized and biologically plausible: that is, its shape should be described by a few intuitive parameters, and it should qualitatively match in-vivo elecrophysiology. While many choices are possible, we define our tuning curve using a linear map (defined by encoders and biases, Eqs 1 and 2) and a rectified linear unit (ReLU). The ReLU neuron is parameterized by an x-intercept (the value of the input current below which a neuron stops firing) and a y-intercept (the neural activity when the input is at its extrema).

We use ReLUs as our target tuning curves for three reasons. First, ReLUs are (arguably) the simplest spiking neurons that still capture an essential neural nonlinearity: the transition from a region (in state space) where the neuron remains (mostly) inactive, to a region where activity increases monotonically (as state inputs change). Second, networks of ReLUs are cheap to simulate but functionally powerful: deep neural networks populated with ReLUs can be trained to perform a wide variety of complex tasks [10]. Third, ReLU parameters are intuitively aligned with the standard NEF technique for generating tuning curve *distributions*. A population of neurons representing a state space input should be sensitive to the whole range of $\mathbf{x}$, otherwise parts of the state space will have low signal-to-noise ratios and the accuracy of the computation will suffer. We must therefore also choose a distribution of target tuning curves that adequately covers $\mathbf{x}$. Previous works have extensively studied how different distributions may effectively represent a state space and dynamically compute functions [4, 11]. One effective distribution is shown in Fig 1: these tuning curves have x-intercepts that vary uniformly across the state space, meaning they are inactive for some values of $\mathbf{x}$ and otherwise have state-dependent activity; and they have y-intercepts (maximum firing rates) that vary uniformly over some range, meaning their spike rates change by different amounts per unit change in $\mathbf{x}$. The combination of these constraints makes it easy to accurately decode an estimate $\hat{\mathbf{x}}$ from the collection of neural activities, and, with ReLU neurons, it is easy to set neural parameters to achieve these constraints.

In contrast to these benefits, the ReLU response function is a poor fit for most electrophysiological data. Notably, as inputs increase, ReLU firing rates increase linearly without bounds, while the activity of biological neurons are constrained to some maximum value, such that firing rates will plateau as neural activity approaches this value. In Sec 4.1, we explore the degree to which biologically implausible target tuning curves affect osNEF's ability to train networks of detailed neurons.

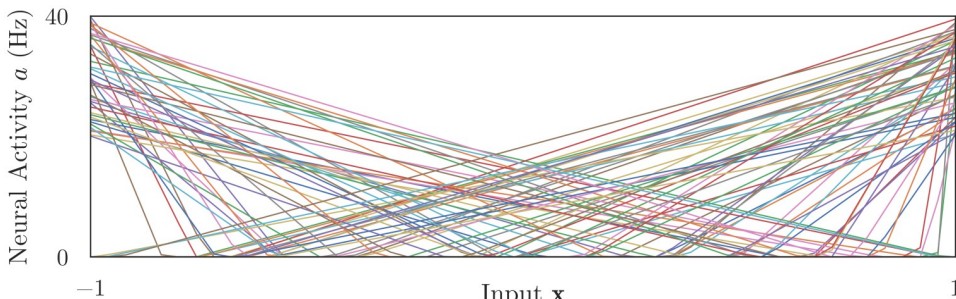

**Fig 1. Target tuning curve distribution.** An example of the distribution of responses for 100 ReLU neurons with x-intercepts in -[1, 1] and y-intercepts in [20, 40].

## 3.2 Online learning rules for encoders and decoders

Given a set of target tuning curves, our first goal is to train synaptic parameters such that, when a population of complex neuron models is simulated within a network, the observed spike rates resemble the rates given by the target tuning curves. The network and procedure used to train these parameters is depicted in Fig 2 and summarized in Table 1. We begin with an input signal $\mathbf{x}(t)$ that is fed into two streams. The top stream of Fig 2 is the "oracle", where the desired transformations of $\mathbf{x}(t)$ are computed analytically (i.e., Eq 6) and used to generate the target spikes (more below), while the bottom stream of Fig 2 is the neural network, where $\mathbf{x}(t)$ is represented by neural spikes and the desired transformations are realized through weighted connections between neural populations. In the oracle stream, $\mathbf{x}(t)$ passes through various filters, represented as boxed $h$s; these operations convolve $\mathbf{x}(t)$ with the filter described in Eq 10 (below). Various state space transformations, which are represented as diamonds containing $f$s (e.g., the identity function $I = f(\mathbf{x}) = \mathbf{x}$), are also applied. Finally, the oracle stream feeds this (filtered, transformed) signal into a population of neurons "tar" that realize the target tuning curves; this generates the neural activities $a^{\mathrm{tar}}(t)$ that osNEF will use to train the neural network proper.

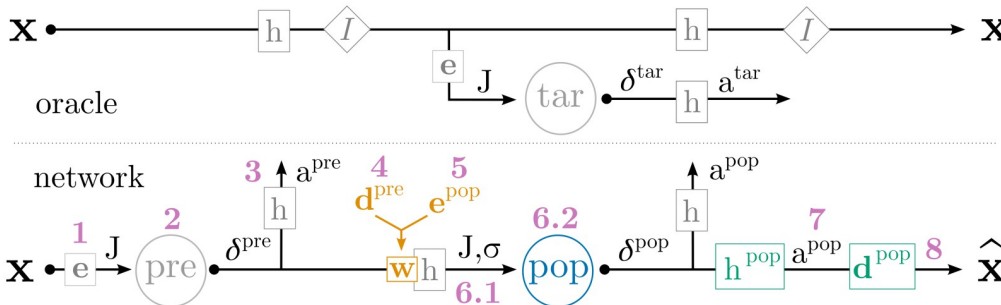

**Fig 2. Network used during osNEF training.** The top half of the figure is the "oracle" stream, where the desired filters and transformations are applied analytically, and where the target activities are generated. The bottom half of the figure is the "network" stream, where synaptic connections realize the desired filters and transformations, and where osNEF training is applied to update the relevant synaptic parameters. Both streams are driven by an input $\mathbf{x}$ (we omit all time-dependencies, such as $\mathbf{x}(t)$ and $J(t)$, for brevity). Arrows represent the signal travelling through each stream. Boxes letters (filters $h$, weights $w$, transformations $I$, and decoders $\mathbf{d}$) indicate mathematical operations being applied to the signal. The resulting quantities (spikes $\delta$, synaptic currents $J$, synaptic conductances $\sigma$, and states $\mathbf{x}$) are shown above the arrow. The pink numbers reference Table 1, which lists the operations that are applied at each step. Circled abbreviations indicate neural populations, which receive synaptic inputs and produce spikes. Coloration indicates ReLU neurons (gray) or detailed neurons (blue), parameters updated by osNEF's online learning rules (orange) or offline synaptic optimization (green), references (pink), and NEF operations (gray).

**Table 1. Summary of equations used during osNEF training, as depicted in Fig 2.**

| Label | Reference | Notes |
|-------|-----------|-------|
| 1 | Eq 1 | converts state space to input current via encoder |
| 2 | Eq 2 | $G$ is a spiking ReLU neuron |
| 3 | Eq 4 | converts discrete spikes to smoothed neural activity |
| 4 | Eq 9 | online update of decoders, which co-determine synaptic weight |
| 5 | Eq 8 | online update of encoders, which co-determine synaptic weight |
| 6.1 | Eq 10 | dynamics for synapse (current- or conductance-based) |
| 6.2 | Sec 3.4 | dynamics for neuron model (may be coupled with synapse) |
| 7 | Sec 3.3 | offline update of synaptic time constants and decoders |
| 8 | Eq 3 | converts neural spikes to state space estimate |

In the bottom stream of Fig 2, the input signal drives a population of preliminary neurons "pre" as per standard NEF encoding: $\mathbf{x}(t)$ is converted to an input current that drives dynamics in the neuron model (Eqs 1 and 2). We arbitrarily chose "pre" to contain ReLU neurons, but any neuron type compatible with standard NEF techniques (e.g., LIF) will do. Neurons in "pre" generate spikes $\delta^{\text{pre}}$ over time, which are smoothed into a real-valued neural activity signal $a^{\text{pre}}(t)$ by convolution with the synaptic filter $h$ according to Eq 4. These activities are then weighted by synaptic weights $w$ and delivered to the population of interest, "pop", which contains detailed neurons. These weights must be trained by osNEF such that the neural activies from "pop", $a^{\text{pop}}(t)$, have the desired tuning properties, introduced in Sec 3.1. Fortunately, the oracle stream provides neural activities $a^{\text{tar}}(t)$ that are guaranteed to reflect the desired tuning properties, by virtue of standard NEF techniques [4, 12], and because the signal that drives "tar" is analogous to the signal that drives "pop". Therefore, if $a^{\text{pop}}(t)$ closely matches $a^{\text{tar}}(t)$, we can say that "pop" has the desired tuning properties.

In order to achieve this, we must train the weights $w$ such that, as spikes from "pre" are weighted, passed through the synapse $h$, and run through the neuron model $G$, $a^{\text{pop}}(t)$ approaches $a^{\text{tar}}(t)$. This is the first objective of osNEF training, and is accomplished using online learning. First, we decompose $w$ into encoders and decoders; for each presynaptic neuron $i$ and postsynaptic neuron $j$, the corresponding entry in the weight matrix is given by

$$w_{ij}^{\text{pre-pop}} = \mathbf{d}_i^{\text{pre}} \cdot \mathbf{e}_{ij}^{\text{pop}}. \tag{7}$$

where $(\cdot)$ is the dot product, $\mathbf{d}_i^{\text{pre}}$ is the $D$-dimensional decoder vector for neuron $i$, and $\mathbf{e}_{ij}^{\text{pop}}$ is the $D$-dimensional vector for the $(i, j)$ neuron pair. osNEF uses two online learning rules, one to update encoders, and another to update decoders.

For encoders, we introduce a new learning rule:

$$\Delta\mathbf{e}_{ij}^{\text{pop}} = \alpha^{\text{e}}\ \text{sign}(\mathbf{d}_i^{\text{pre}})\ a_i^{\text{pre}}\ (a_j^{\text{pop}} - a_k^{\text{tar}}) \tag{8}$$

where $\alpha^{\text{e}}$ is the encoder learning rate, $\text{sign}(\mathbf{d}^{\text{pre}})$ is the elementwise sign of the presynaptic decoder, and $(i, j, k)$ are neuron indices for the presynaptic, postsynaptic, and target neuron populations, respectively. The rule is supervised in the sense that the target activities are provided in real time as the oracle drives "tar", and the difference between the current and target activities drives the update. The rule also utilizes information about both presyaptic and postsynaptic activities, making it Hebbian. Note that Eqs 7 and 8 redefine encoders as a tensor, indexed over $i$, $j$, and $D$, rather than a matrix over $j$ and $D$, as is standard in the NEF (Eq 2).

For decoders, we use the Prescribed Error Sensitivity (PES) learning rule [13], an online, error-driven learning rule that is frequently used to train NEF networks:

$$\Delta \mathbf{d}^{\text{pre}} = \frac{\alpha^{\text{d}}}{N_i} \; a^{\text{pre}}(t) \; (\hat{\mathbf{x}}(t) - \mathbf{x}(t)) \tag{9}$$

where $\alpha^{\text{d}}$ is the decoder learning rate, $N_i$ is the number of presynaptic neurons, $a^{\text{pre}}(t)$ are the filtered activities from presynaptic neurons, $\mathbf{x}(t)$ is the state space target, and $\hat{\mathbf{x}}(t)$ is the decoded estimate (Eq 3 with $a^{\text{pre}}(t)$ and $\mathbf{d}^{\text{pre}}$). Previous work has shown that this learning rule is capable of learning decoders to compute a wide range of functions [12]. Note that, although Eq 7 decomposes weights into encoders and decoders, only the combined weights are actually used to transform signals within the network: encoders and decoders are theoretical tools used to analyse the relationship between spike space and state space and to facilitate training, but synaptic transmission between populations of detailed neurons is governed only by weight.

### 3.3 Optimizing synaptic time constants

Continuing on the bottom stream of Fig 2, the spikes generated by "pre" must pass through synapses that (a) convert $\delta^{\text{pre}}(t)$ to the state variables used in the neuron model, and (b) apply the weights $w$ that realize the target transformations. For our simpler neuron models, synapses deliver current to the cells, which directly affects the cell's voltage. For our complex neuron model, synapses update the conductance parameters in the relevant sections of the cell, which then influence transmembrane currents that govern voltage change. In both cases, we assume that the synapse is a second-order lowpass, or double-exponential, filter, whose transfer function is

$$h(s) = \frac{1}{(\tau_{\text{rise}} \; s + 1) \; (\tau_{\text{fall}} \; s + 1)} \tag{10}$$

where $\tau_{\text{rise}}$ and $\tau_{\text{fall}}$ are time constants and $s$ is in the Laplace domain. Whenever a synapse receives a spike, it updates the postsynaptic cell's input current (or conductance) by an amount proportional to the synapse's dynamical state and its weight. To ensure that our decoded estimates align with the signals being transmitted in the network, we also use the double exponential filter to estimate neural activities from neuron spiking outputs (Eq 4).

This leaves the question of how to choose $\tau_{\text{rise}}$ and $\tau_{\text{fall}}$. When smoothing spikes for the purpose of encoder learning (Eq 8) or synapsing from "pre" onto "pop", the choice of time constant makes little difference, so long as it sufficiently smooths spike noise (e.g., $\tau_{\text{fall}} > 10$ms). However, as we show in the Sec 4.1, the choice of time constants makes a significant difference when (a) decoding $\hat{\mathbf{x}}(t)$ from "pop", or (b) connecting one population of detailed neurons to another. We could choose $\tau$ values based on the effective time constants of biological neurotransmitters, but it is unclear whether one set of parameters would be appropriate for the variety of neuron types and networks that we investigate. To resolve this problem, osNEF uses a novel offline optimization procedure that finds appropriate values for these constants given spiking data from the simulated network itself. The optimization procedure is as follows:

1. Simulate the network with input $\mathbf{x}(t)$, record neural spikes $\delta^{\text{pop}}(t)$, and specify the target function $f(\mathbf{x}(t))$.

2. Choose a random $\tau_{\text{rise}}$ and $\tau_{\text{fall}}$, filter the spikes to calculate $a^{\text{pop}}(t)$, and use least-squares to compute decoders $\mathbf{d}^{\text{pop}}$ for estimating the function.

3. Calculate the error between this estimate and the ground truth by computing the RMSE between $f(\mathbf{x}(t))$ and $\hat{f}(\mathbf{x}(t))$.

4. Repeat Steps 2 and 3, using the optimization package Hyperopt [14] to search the space of possible time constants with the objective of minimizing the error.

Returning to Fig 2, this procedure is used to train the filters and decoders for "pop", $h^{pop}$ and $\mathbf{d}^{pop}$. Filtering spikes from "pop" produces neural activities $a^{pop}(t)$, from which we can decode an estimate $\hat{\mathbf{x}}(t)$ according to Eq 3. This completes the bottom stream, which shows how a state space input may be translated into neural spikes, transformed to realize particular mathematical operations via synaptic connections, then translated back to a state space estimate.

## 3.4 Neuron models

To test whether osNEF is capable of producing functional networks that contain neuron models of varying complexity, we investigate four neuron models: the LIF neuron, the Izhikevich neuron, the Wilson neuron, and a Layer V Pyramidal Cell reconstruction.

The LIF neuron model is a point neuron that approximates the membrane dynamics preceding and following an action potential. Although the resulting voltage traces do not quantitatively align with electrophysiological recordings, the LIF neuron does capture key features of neural behavior, namely integration of inputs, leak towards a resting potential, reset following a spike, and a refractory period. It is also extremely cheap to simulate, as voltage dynamics are governed by a single equation. As such, LIF neurons are widely used in simulations that seek to balance biological realism, computational scalability, and analytical tractability (see [4]).

Although LIF neurons are fast and functional, they do not quantitatively capture the dynamics of membrane potential. The Izhikevich neuron model [15] is another simple neuron model that captures a wider variety of spiking behavior characteristic of biological neurons. The model has only four free parameters and two state variables, but certain configurations of these parameters may produce regular spiking, intrinsic bursting, fast spiking, chattering, and many more interesting dynamics. As such, this neuron model is useful in networks where both scallability and electrophysiological realism are important.

While the LIF and Izhikevich neurons are useful and computationally cheap neuron models, they do not simulate the action potential in detail, instead using hand-crafted reset mechanisms when the cell's voltage crosses a fixed spike threshold. For our third neuron model, we chose an intermediate-complexity neuron developed by Wilson [16] that extends the Fitz-Hugo-Nagumo equations [17, 18] to incorporate electrophysiological detail, including Ohm's Law and equilibrium potentials of four ionic currents in neocortical neurons ($I_k$, $I_{Na}$, $I_T$, $I_{AHP}$). The resulting model consists of three coupled ODEs representing voltage, conductance, and recovery, can generate realistic action potentials, and naturally produces adaptation, bursting, and other neocortical behaviors [16]. Due to the lower number and cubic dynamics of the underlying equations, simulation is still relatively fast, but a smaller timestep is required to avoid numerical errors.

When describing electrophysiology in detail, the most widely-used formalism is Hodgkin-Huxley, which we use for our final neuron model. Reproduced from Durstewitz, Seamans, and Sejnowski [19], this model is an anatomically-detailed reconstruction of pyramidal neurons that includes four compartments (soma, proximal-, distal-, and basal-dendrites) and six ionic currents (two for sodium, three for potassium, and one for calcium). The Durstewitz reconstruction accurately reproduces electrophysiological recordings from layer-V intrinsically-bursting pyramidal neurons in rat PFC, cells that are known to be active during the delay

period of working memory tasks. This neuron model is implemented in NEURON and uses conductance-based synapses, distributed randomly on the three dendritic compartments.

## 4 Results

To demonstrate that osNEF is capable of training cognitively-useful neural networks built from a variety of neuron models, we divide our results into three sections: representation, computation, and application. First, we show that osNEF produces populations of neurons with the desired tuning properties, and demonstrate that their spiking responses *represent* the target signal. These results indicate that an input signal may be encoded and decoded effectively by a single population of biologically detailed neurons using weights and filters trained using osNEF. Next, we simulate networks containing multiple populations of biologically detailed neurons, and show that the synaptic connections between them *compute* specific functions and exhibit the target dynamics. These results show that osNEF combines online learning and offline optimization to realize encoding, decoding, and dynamics in a single synaptic process that occurs between populations of detailed neurons. Finally, we *apply* these techniques to train a biologically-detailed model that performs a working memory task. These results suggest that osNEF is a useful tool for researchers who wish to build, manipulate, and validate biologically detailed simulations of cognitive systems.

### 4.1 Representation

First, we show that encoding and decoding are possible with a single population of detailed neurons. We simulated four networks using the architecture shown in Fig 2; each network had one LIF, Izhikevich, Wilson, or Pyramidal neuron in the "pop" population, and one ReLU neuron in the "target" population. The networks were trained using the online learning rules described in Sec 3.2, then presented with a novel input during testing. The top and middle panels of Fig 3 show the input signal $\mathbf{x}(t)$ (smoothed, 1Hz band-limited white-noise) and neural activities over time, respectively. In response to this input, all four neurons dynamically exhibit spiking activity that closely aligns with the spiking activity of the target ReLU neuron. The bottom panel of Fig 3 shows the observed tuning curves calculated from these data: for each timestep in the simulation, we found the state space value of the input signal $\mathbf{x}(t')$ and recorded the smoothed neural activity $a(t')$ at that time. We divide the state space into 21 equally-sized regions (or bins), then associate each $a(t')$ with the appropriate bin. Finally, we plot the mean and 95% confidence interval of neural activity for each bin.

Examining the observed tuning curves in Fig 3, we see that all four trained neurons have x- and y-intercepts that closely align with the intercepts of the target ReLU neuron. However, the y-intercept of the ReLU neuron is somewhat higher than the trained neurons. This result is expected: ReLU activity will continue to ramp without bounds, while more realistic neuron models will exhibit plateauing activity. Although this is, generally speaking, a danger of using ReLUs (or other unrealistic neuron models) as spike space targets, it is not problematic in our simulations for two reasons. First, for the state spaces that we simulate, and for the maximum firing rates we target, the differences between the ReLU and trained tuning curves are minimal: in Fig 3, ReLU activities only exceed the trained activities for extreme inputs ($\mathbf{x} > 0.8$), and these differences fall within the observed confidence intervals. Second, because osNEF training minimizes the differences between the spiking activity of a trained neuron and a target neuron, trained neurons will often match the "physiologically plausible" features of a target tuning curve but fail to match any "physiologically implausible" features. This tendency can also be observed in Fig 3: for most inputs ($\mathbf{x} < 0.8$), our trained neurons qualitatively match the inactive and linear segments of the target ReLU curve, but for extreme inputs ($\mathbf{x} > 0.8$), our trained

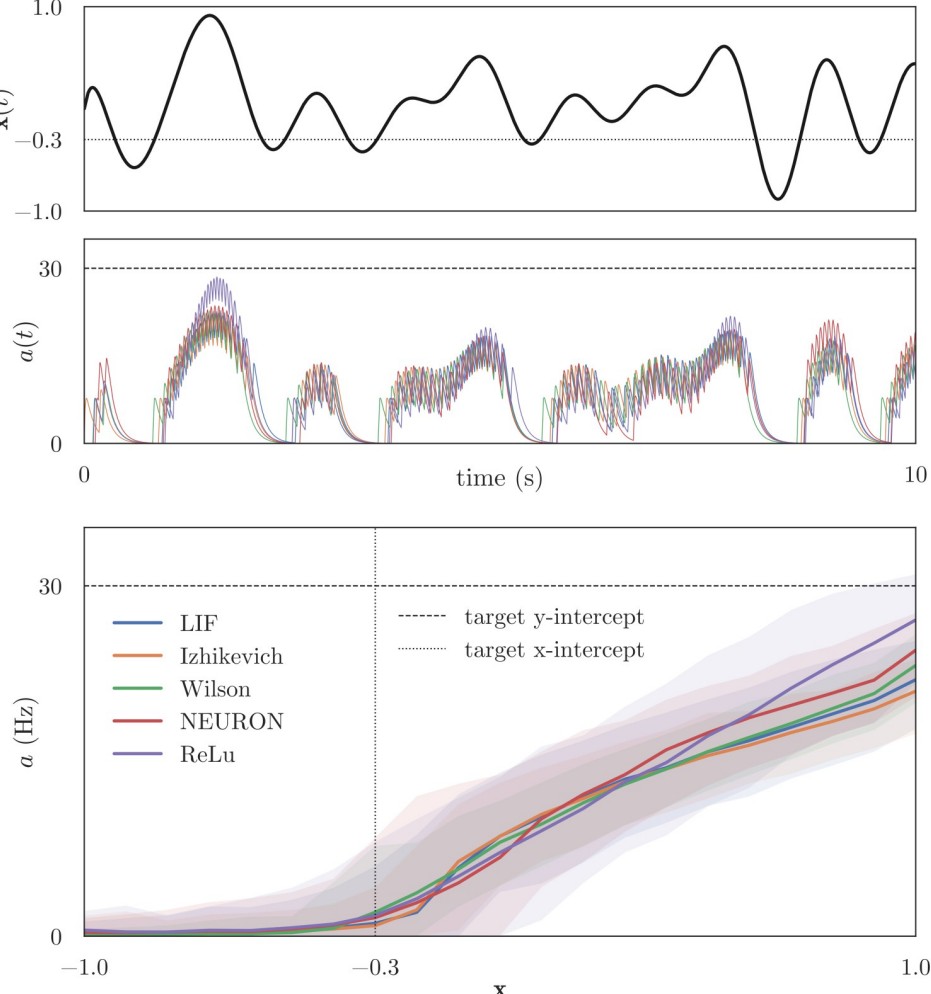

**Fig 3. Encoding and tuning curves.** The top panel shows the input signal and the target x-intercept, the state space value at which our neurons should begin spiking. The middle panel shows neural activity over time and the target y-intercept, the desired activity of our neurons when the input is at its maximum value. The bottom panel shows the tuning curves derived from these data. All four neuron models exhibit minimal spiking activity when the input is below the target x-intercept; neural activities also increase as the value of the input increases, up to the target y-intercept. Shaded error regions indicate 95% confidence intervals for smoothed activity, and demonstrate that all simulated neuron models have a natural variation in firing rate for any given state space value **x**. The significant overlap between the four trained neuron models and the target (ReLU) activities shows the success of our online learning rule.

neurons begin to plateau, failing to match the target tuning curve *but* retaining realistic response properties. Ultimately, any target tuning curve will respond differently than the neuron model being trained. By choosing a target tuning curve that is manifestly implausible in some respects, then showing that our trained neurons still achieved the desired response properties, we demonstrate that osNEF does not require perfect selection of target tuning curves. More importantly, Fig 3 demonstrates that osNEF trains encoders effectively: our learning rules produce dynamic activities that closely resemble the target tuning curves for all four neuron models, despite significant differences in neural complexity and cellular dynamics.

To investigate decoding, we simulated four populations that contained 100 LIF, Izhikevich, Wilson, or Pyramidal neurons, and used osNEF to train encoders, decoders, and readout filters

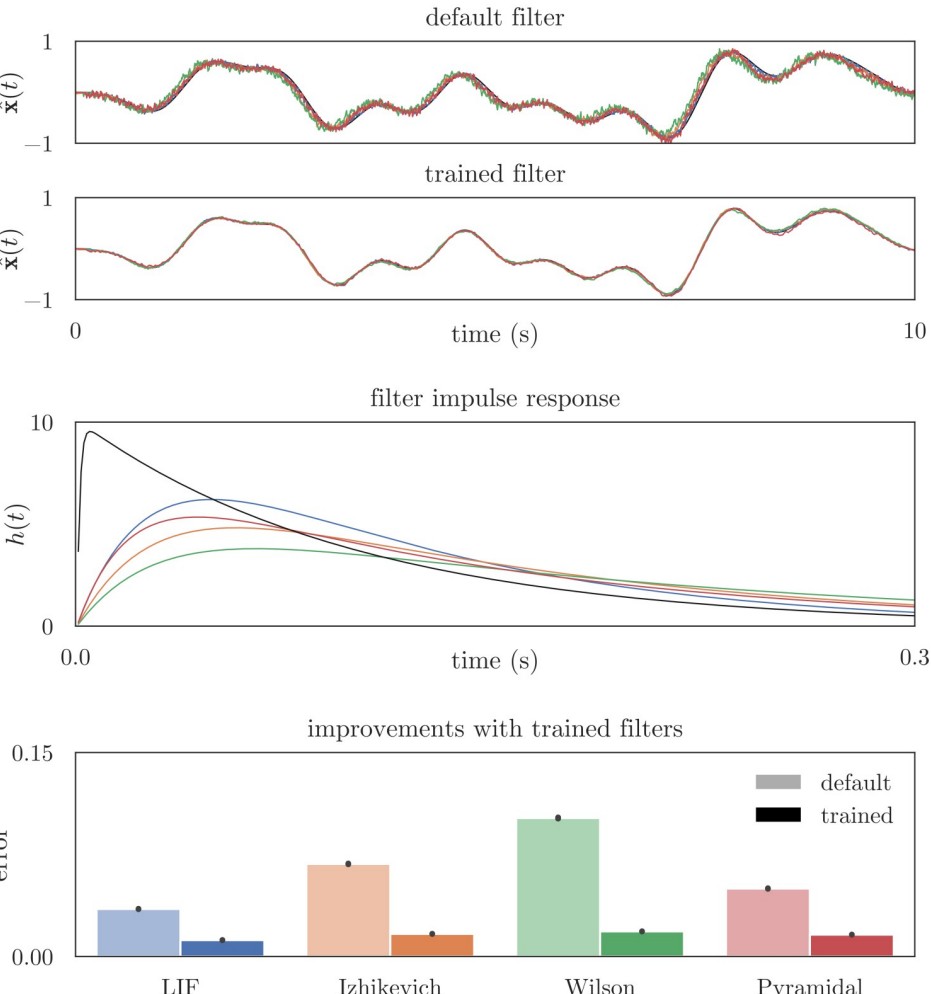

**Fig 4. Decoding and readout filters.** Nonlinear dynamics within complex neuron models leads to systematic decoding error if a default filter is used to smooth the spikes. Optimizing a filter's time constants accounts for this problem and reduces spike noise, leading to highly accurate estimates across all neuron models. The top two panels show the target values and the state space estimates, which are decoded from the activities of 100 LIF, Izhikevich, Wilson, or Pyramidal neurons in "pop"; spikes are smoothed using either the default filter (first panel) or the osNEF-trained filter (second panel). The third panel confirms that the RMSE between state space targets and decoded estimates (averaged across 10 simulations with unique inputs) are significantly lower when using the trained filter. The fourth panel shows the impulse response functions.

for each population. An interesting problem arose when we used Eqs 3 and 4 to decode the neural activity and estimate the state space representation in "pop". When we used a default filter to smooth the spike trains ($\tau_{\text{rise}}$ = 1ms and $\tau_{\text{fall}}$ = 100ms), our decoded estimates $\hat{\mathbf{x}}(t)$ were often phase-shifted to the left of the target $\mathbf{x}(t)$, leading to systemic error, as shown in Fig 4. The phase shift is more pronounced in neurons with greater observed spike adaptation or variance in interspike intervals, most notably the Wilson neuron. To account for this phase shift and decode a better estimate from the neural spikes, we used our synaptic optimization to find better readout filters for each of the four networks. The optimization produced filters with longer time constants, which effectively (a) delays the signal and negates the leftward phase shift, and (b) smooths noisy spike trains to recover a more accurate estimate of the input signal. Panel one and two of Fig 4 show the decoded estimate when calculating $\hat{\mathbf{x}}(t)$ using the default

filter versus the trained filter. Panel three shows the impulse responses of the trained filters, while panel four compares the RMSE between $\hat{\mathbf{x}}(t)$ and $\mathbf{x}(t)$, when filtering with the default versus trained filters, across 10 input signals. The gains in accuracy with the trailed filter are substantial, and demonstrate that osNEF is capable of accurately decoding state space signals from the activities of nonlinear, adaptive neurons. In the Sec 4.2, we report results from networks whose synaptic filters have been trained using the above methods; see S1 Appendix for a table of the optimized time constants and a stability analysis of the Hyperopt parameter search.

## 4.2 Computation

Having established that encoders, decoders, and filters may be used to translate between the spike space and the state space in a single neural population, we now apply our method to train neural networks that compute cognitively-useful functions using the weighted synaptic connections between two (or more) populations of detailed neurons. The simplest network is a communication channel, which simply computes the identity function. The network architecture is shown in Fig 5; the target function is computed between detailed neuron populations "pop$_1$" and "pop$_2$". The target function is

$$f(\mathbf{u}, \mathbf{x}) = \mathbf{u}, \tag{11}$$

where $\mathbf{u}(t)$ is the input signal and $\mathbf{x}(t)$ is the state space representation. A functioning communication channel ensures that information can be relayed between components of a cognitive system without significant loss, or be decoded by muscle effectors to implement behavior. Fig 6 shows that osNEF successfully trains encoders, decoders, and time constants that preserve the input signal: the target signal can be reliably decoded from "pop$_2$" with very low error for all four neuron models. While Fig 3 showed that encoder learning leads to representative spiking activity, and Fig 4 showed that this activity may be decoded to retrieve the signal, Fig 6 shows that encoding and decoding may be combined into a single step via neural connection weights. See S2 Appendix for an investigation of how external noise affects performance when computing feedforward functions.

We also constructed a second network that multiplies two scalars to produce a new scalar:

$$f(\mathbf{u}, \mathbf{x}) = \mathbf{u}_1 \odot \mathbf{u}_2, \tag{12}$$

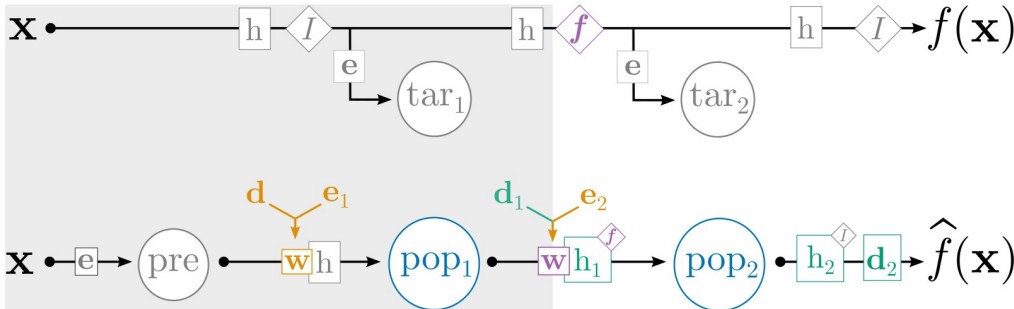

**Fig 5. Network architecture for computing feedforward functions, including the identity function and multiplication of two input scalars.** This network extends the training network in Fig 2, represented by components with the gray background, by including an additional detailed neuron population "pop$_2$" and the corresponding oracle components. With this architecture, we can compute the feedforward function $f(\mathbf{x})$ on the connection between "pop$_1$" and "pop$_2$" by using osNEF to train the synaptic parameters $\mathbf{d}_1$, $\mathbf{e}_2$, and $h_1$. As before, coloration indicates ReLU neurons (gray) or detailed neurons (blue), synaptic parameters trained by online learning (orange) or offline optimization (green), NEF computations (gray), and finally the new components involved in the calculation of $f(\mathbf{x})$ (purple).

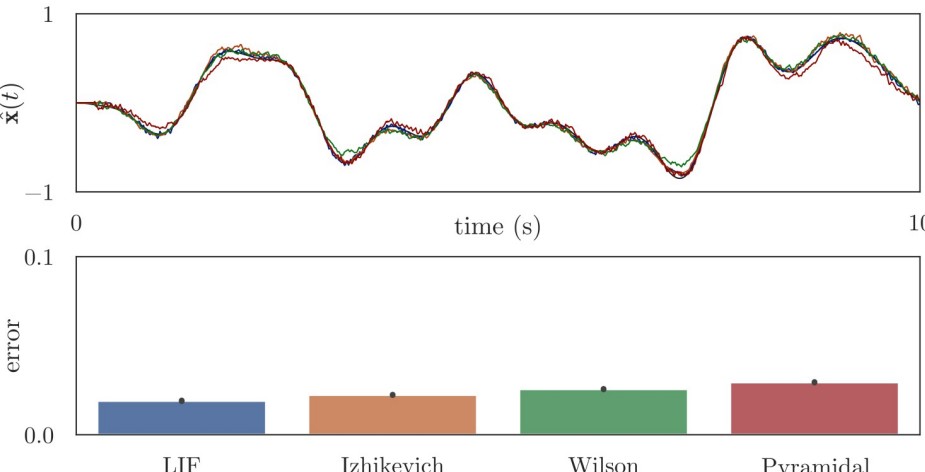

**Fig 6. Computing the identity function, Eq 11.** Using the network architecture in Fig 5, we initialize neural populations "pop₁" and "pop₂" with 100 detailed neurons, then use osNEF to train encoders, decoders, and synaptic filters. The connection between "pop₁" and "pop₂" is trained to compute the identity function, such that "pop₂" represents the same information as "pop₁". The top plot shows the state space target and the decoded estimates from "pop₂", and the bottom plot shows the mean error (RMSE) between this estimate and the target across 10 simulations with unique input signals.

where $\odot$ is the element-wise product. Multiplication is a computational primitive that may be used in a wide variety of cognitive systems to transform simpler representations into more complex ones. For example, binding operations may be used to associate sensory representations with recalled memories or emotional reactions, producing a combined representation that may be semantically processed by downstream systems. Although the precise mechanisms for binding in the brain remain uncertain, many neural implementations of binding rely on multiplying elements of the represented vectors [3]. Fig 7 shows the state estimate decoded

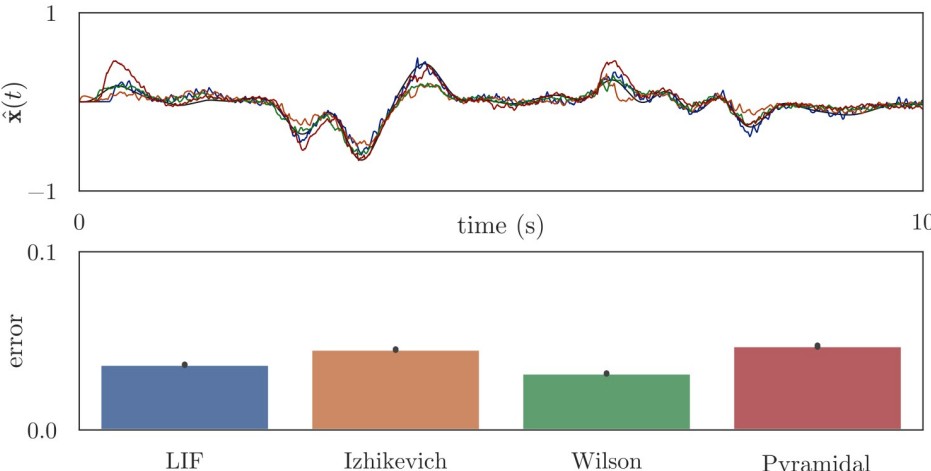

**Fig 7. Computing the product of two unique input signals, Eq 12.** Using the network architecture in Fig 5, we initialize neural populations "pop₁" and "pop₂" with 100 detailed neurons, then use osNEF to train encoders, decoders, and synaptic filters. The connection between "pop₁" and "pop₂" is trained to multiply two scalars: "pop₁" represents the two scalars, and "pop₂" should represent their product. The top plot shows the state space target and the decoded estimates from "pop₂", and the bottom plot shows the mean error between this estimate and the target across 10 simulations with unique input signals.

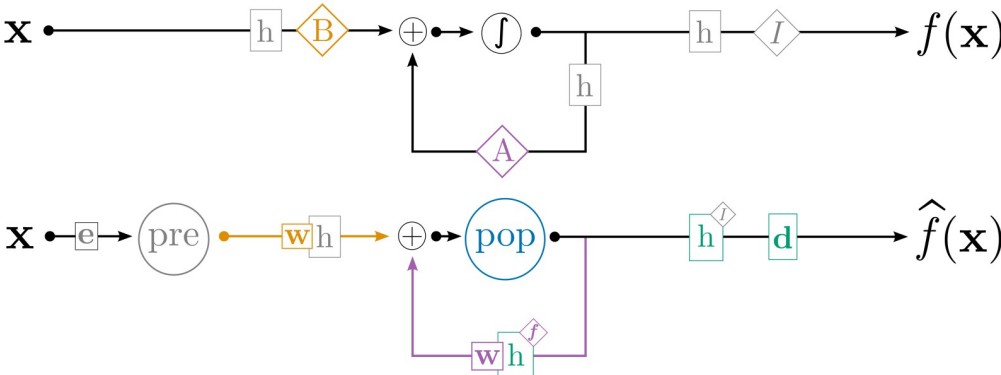

**Fig 8. Network architecture for recurrent networks.** Orange components indicate the feedforward computation of $\dot{\mathbf{x}} = B\mathbf{u}$, and purple components indicate the recurrent computation of $\dot{\mathbf{x}} = A\mathbf{x}$; together they implement Eq 6. While this network is used at test-time, an "unrolled" version similar to Fig 5 is used during training. As such, we remove reference to "tar", and to the decoders and encoders composing $w$, in this figure.

from the final population for each of our neuron models. Errors are higher than in Fig 6, as expected given the increased difficulty of the computation. Still, all four neuron models perform well overall.

Many biological systems include recurrent connections, allowing the currently represented state to directly affect future states and permitting a wider variety of network dynamics, including rhythmic oscillations and working memory. A network that includes feedforward and feedback components may implement any dynamical systems described by Eq 6. The network architecture we use for simulating recurrent networks is shown in Fig 8: feedforward computation of $\dot{\mathbf{x}} = B\mathbf{u}$ occurs on the connection between "pre" and "pop", while feedback computation of $\dot{\mathbf{x}} = A\mathbf{x}$ occurs on the recurrent connection on "pop". To train the network, we reuse the architecture in Fig 5, but set "pop$_1$" and "pop$_2$" to be identical populations of detailed neurons computing the target recurrent function $\dot{\mathbf{x}} = A\mathbf{x}$. In doing so, we effectively "unroll" the recurrence, as is common when training recurrent neural networks with backpropagation, but still use osNEF to train network parameters given a dynamic input signal.

The first recurrent system we investigate is a simple harmonic oscillator, that is, a two-dimensional oscillator with frequency $\omega$.

$$\dot{\mathbf{x}} = \begin{bmatrix} 0 & -\omega \\ \omega & 0 \end{bmatrix} \mathbf{x} \tag{13}$$

This oscillator is a classic example of central pattern generation: after a brief kick, the system should maintain oscillatory dynamics without external input, which may be used to drive rhythmic behavior in the body, or provide carrier signals that may be modulated by downstream cognitive systems. We arbitrarily chose $\omega = 2\pi$ as our target frequency. Fig 9 shows the network dynamics after a square-wave pulse (0.1s) is used to kick the system. All four neuron types quickly settle into stable harmonic oscillation with frequency approximating the target $\omega$, and these oscillations persist for 100 seconds. Because the networks invariably oscillate at a frequency that differs slightly from $\omega$, a naive calculation of RMSE between $\mathbf{x}$ and the decoded $\hat{\mathbf{x}}$ is a poor metric of the system's stability. To account for this, we report two error values: we first fit a sinusoid of the form $a \sin(bt + c) + d$ to $\hat{\mathbf{x}}$, then report (a) the RMSE between this sinusoid and the neural estimate, and (b) the normalized frequency error $(b - \omega)/\omega$.

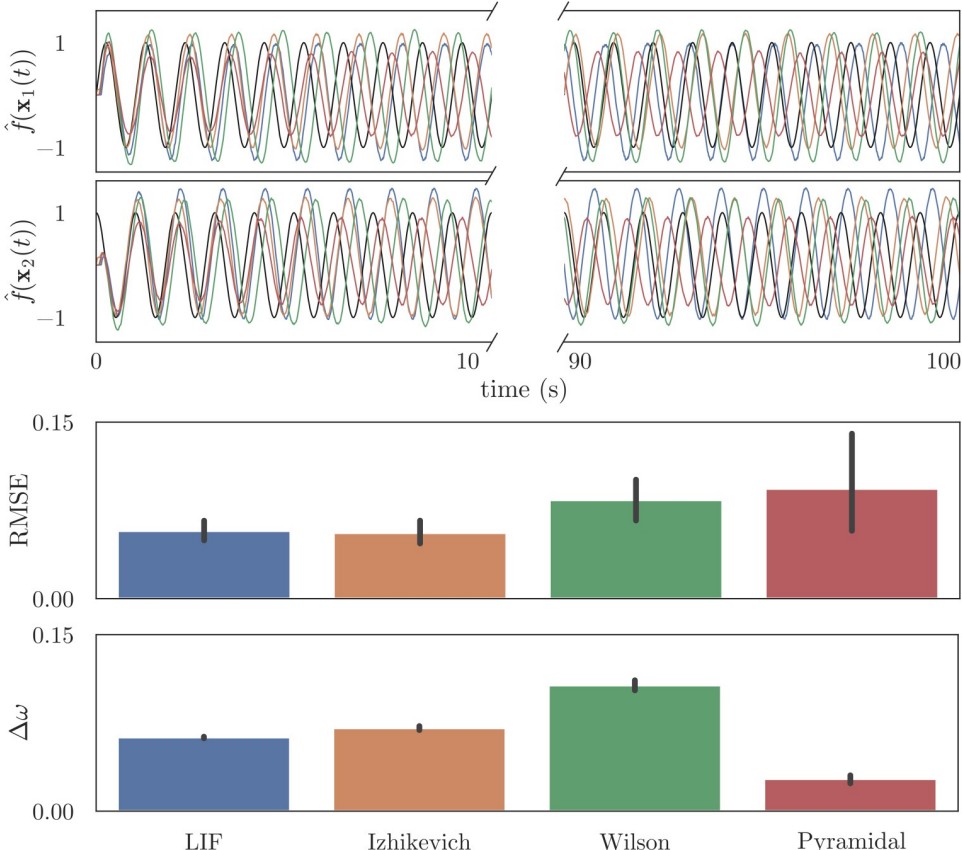

**Fig 9. Implementing a simple harmonic oscillator (Eq 13) using a recurrent connection.** Using the network architecture in Fig 5, we initialize neural populations "pop₁" and "pop₂" with 100 detailed neurons, then use osNEF to train encoders, decoders, and synaptic filters. The connection between "pop₁" and "pop₂" is trained to compute Eq 13. The weights and synapses from this trained model are then substituted into a testing network, shown in Fig 8. The top panel shows the state space target and the decoded estimates from "pop₂", with a break in the x-axis to show that oscillations remain stable over 100 seconds. The bottom panel show the mean error between this estimate and a best-fit sinuoid, as well as the frequency error between this best-fit sinuoid and the target frequency $\omega = 2\pi$.

The second recurrent system we investigate is an integrator, a system which continually adds a feedforward input to a remembered representation of its current value:

$$\dot{\mathbf{x}} = \mathbf{u}. \tag{14}$$

This dynamical system requires a neural network to continuously combine feedforward and recurrent signals, an important operation for working memory. We use the network in Fig 10 to realize such a memory, which we refer to as a "gated difference memory" [20]. The purpose of this network is to (a) read an input value and represent that value using neural activities in a recurrent population, and (b) continue to represent that value once the input has been removed. Such a network may perform working memory tasks such as the delayed response task (DRT), in which an animal must remember the 2D location of a briefly-presented visual cue for a short period of time before recalling its location [21]. In our network, the feedforward connection from "pre" to "pop" passes to the memory a two-dimensional value (representing, for example, a visual cue), while the recurrent connection on "pop" maintains the represented cue location once the input has been removed. This is the core "memory" component. The second component is the "difference" component, which ensures that the

**Fig 10. Network architecture for the gated difference memory.** This system loads and stores a two-dimensional value in a working memory; when the "gate" signal is on (closed), the system maintains its current value through recurrent activity, and when the "gate" is off (open), the system replaces its current representation with the input value. Gray circles are LIF populations while blue circles are detailed neuron populations. The green connection is trained by osNEF to compute $f(\mathbf{x}) = \mathbf{x}$, while the purple connection is trained to compute $f(\mathbf{x}) = -\mathbf{x}$. The orange connection directly inhibits neurons in "diff" using fixed negative weights.

value represented in "pop" approaches the cue value represented in "pre". Because the recurrent connection continuously computes the identity function $f(\mathbf{x}) = \mathbf{x}$, maintaining whatever 2D value is currently represented, and the feedforward connection continuously adds the 2D value of the perceived cue to the representation in "pop", it is possible for a naive integrator to "overshoot" the target value if the cue is presented for an extended duration. To prevent this, we add an intermediary population "diff" between "pre" and "pop". This population receives the feedforward signal from "pre" and transmits feedforward to "pop", acting as a simple pass-through. However, it also receives a feedback connection from "pop" that computes the negative of the identity function $f(\mathbf{x}) = -\mathbf{x}$. As a result, the value represented in "diff" is equal to the cue's value minus the integrator's estimate; when this estimate becomes equal to the cue's value, "diff" should represent zero, and the representation in "pop" should stabilize at the target value. Finally, the "gate" allows the network to ignore the input and simply retain its current representation. When the visual cue is removed, we treat its absence as a secondary input to the system, which activates a population of neurons "inh" that inhibits "diff", preventing any further update of the representation in "pop". To recall an estimate of the remembered cue's location, we simply decode the neural activities in "pop" with the identity function.

Fig 11 shows the DRT performance of our network for each neuron type. In each trial, we present a cue to the network for 1s, then close the gate and record the decoded estimate from "pop" over time, computing the RMSE between this value and the original input over a 10s delay period. The cues are distributed evenly around the unit circle across our 10 trials, and we report the RMSE averaged across the 10s delay period and the 10 trials. Once the input is removed, the value represented in "pop" must be maintained through the recurrent connection; over time, noise inevitably causes this system to drift away from the target value, leading to imperfect recall. However, for most of the presented cues, the network settles on an attractor that is proximate to the target value, leading to reasonable error rates across all neuron types.

## 4.3 Application

We now demonstrate how osNEF may be used to build a simple cognitive model out of biologically-detailed components. As above, we use a neural integrator to model a DRT, in which an animal must read information from an external signal, remember that information for a period of time once the signal has been removed, then recall that information. Numerous researchers use DRTs to investigate the neural basis for working memory [19, 21–23], and

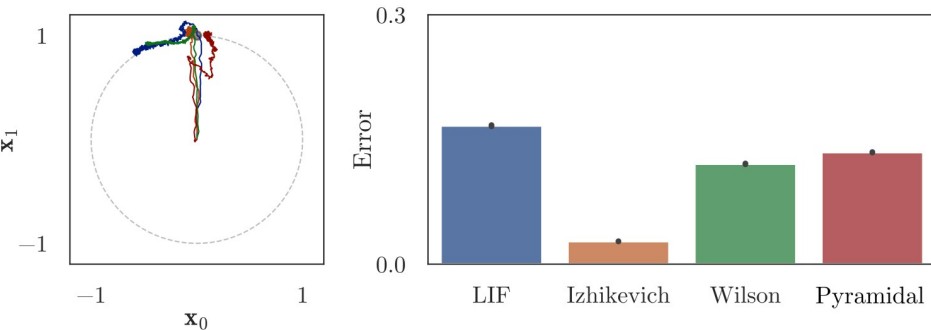

**Fig 11. Implementing a gated difference memory using a combination of feedfoward, recurrent, feedback, and inhibitory connections.** The left panel shows the estimate decoded from "pop" as a trajectory in $(x, y)$ space: as the cue is presented, the estimate travels from the origin $(t = 0)$ to the cue's location, which lies somewhere on the unit circle. At $t = 1$ the cue is removed, and the system must rely on its recurrent dynamics to maintain a stable estimate of the cue's location. We observe minimal drift in the decoded trajectories for most cue locations, indicating that our memories are fairly stable over time. The right panel shows the Euclidean distance between the decoded estimate and the cue's true location, averaged over a 10s delay period and over 10 cue locations, for each neuron model.

neural integrators have been used to model working memory in larger cognitive models that reproduce human behavioral data [3, 24, 25].

We extend the gated working memory network described in Sec 4.2 by enforcing additional biological constraints and adding an associative memory with winner-take-all (WTA) dynamics to select actions, as shown in Fig 12. As above, we use the Pyramidal cell model proposed by Durstewitz et al., as this cell reconstruction was explicitly designed to simulate pyramidal neurons with delay-period activity in working memory tasks [19]. We also use Durstewitz's (a) cellular reconstruction of inhibitory interneurons; (b) conductance-based synapse models for GABA and AMPA; (c) conductance-based, voltage-gated synapse model for NMDA; and (d) biophysical simulation of Dopamine (DA). In these models, DA affects the activation threshold for the persistent $Na^+$ current, the conductance of the slowly inactivating $K^+$ current and high-voltage-activated $Ca^{2+}$ current, and the magnitude of NMDA, AMPA, and GABA synaptic conductances. The parameter values were taken directly from the Durstewitz's original source code and were not modified to improve the performance of our model. As in Secs 4.1–4.2, we use osNEF to train synaptic weights, resulting in dense connectivity that includes both

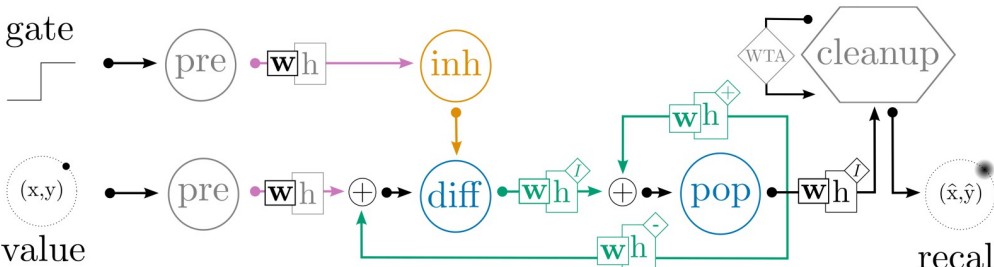

**Fig 12. Architecture for the biologically-constrained DRT neural network.** This network extends Fig 10 by (a) replacing "inh" with a population of detailed inhibitory interneurons, (b) adding a "cleanup" network that uses WTA competition to find the cue location that best resembles the recalled location from "pop" (see [26] for a detailed description), and (c) replacing all connections to/from detailed neurons with conductance-based AMPA, GABA, or (voltage-gated) NMDA synapses. Grey populations contain LIF neurons, orange populations are interneurons, and blue populations are pyramidal cells. Pink connections use AMPA synapses, orange connections GABA synapses, and green connections use NMDA synapses.

excitatory and inhibitory synapses between pyramidal cells and interneurons. Unlike in Secs 4.1–4.2, we do *not* use the osNEF to optimize synaptic time constants, since the time constants of AMPA, NMDA, and GABA synapses are fixed to their biological values.

Our cognitive neural network performs a two-dimensional DRT. The network is first presented with an input cue, which represents the $(x, y)$-location of a target point on a visual screen, for 1s. The cue is removed, followed by a delay period, during which the network continuously reports its remembered estimate of the cue's location. This estimate is sent to an associative memory, which stores the possible true locations of the target; the associative memory compares the current estimate of the cue's location to these targets and outputs the target vector with the greatest similarity, effectively acting as a cleanup operation for the remembered location [27]. We classify a response as "correct" if the output of the cleanup memory falls within a certain (Euclidean) distance of the target cue, and measure the percentage of correct responses as a function of the delay period length. To ensure robust results, we (a) train ten unique networks, whose tuning curve distributions and training signals are seeded with different random numbers, and (b) test each network's recall accuracy for eight cues with $(x, y)$ locations distributed evenly around the unit circle.

Fig 13 reports the forgetting curves for the biologically-detailed network, plotting the percentage of correct responses given by the model as a function of the delay length. We use Scipy's `curve_fit` function to estimate parameters for an exponential forgetting curve ($y(t) = B \exp(-t/\tau)$) that fits the simulated data from each network, then report the range, mean, and median of these parameters across the networks. When preprocessing the data, we observed that the maximum accuracy achieved by the network often occurred 500–2000ms after the cue was removed, presumably as a result of the long NMDA time constants and recurrent dynamics within the network. To avoid numerical errors when using `curve_fit`, we set $B$ equal to this accuracy (for each network) and assumed an exponential rate of forgetting beyond this value; we also transformed $\tau$ into a "half-life" time constant by multiplying with ln2. This resulted in best fit parameters $B$ ranging from 58% to 100%, with mean 89% and median 95%; and $\tau_{half}$ ranging from $2.7 - 28.0$s, with mean 9.6s and median 8.6s. The exponential curve is a good fit for the simulated data, a result consistent with numerous studies showing that animal performance on working memory tasks declines exponentially as the length of the delay interval increases [28, 29].

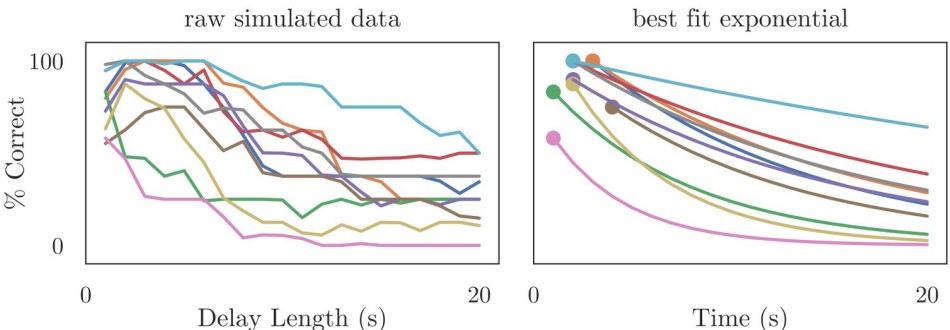

**Fig 13. Mnemonic performance of the biologically-detailed cognitive network and the best fit exponential forgetting curve.** We trained the network show in Fig 12 using osNEF to produce a gated difference memory, as described in Sec 4.2 and Fig 10. Rather than average the error over time, as we did in the right panel of Fig 11, we plotted error (percentage of correct responses over 8 cue locations) as a function of time. We repeated this training and testing procedure for ten networks seeds, treating each network as an individual "participant" performing this task, then fit an exponential function to each network's forgetting curve. From these fitted curves, we obtained parameters for baseline performance and performance half-life, which we compared with the empirical data shown in Fig 14.

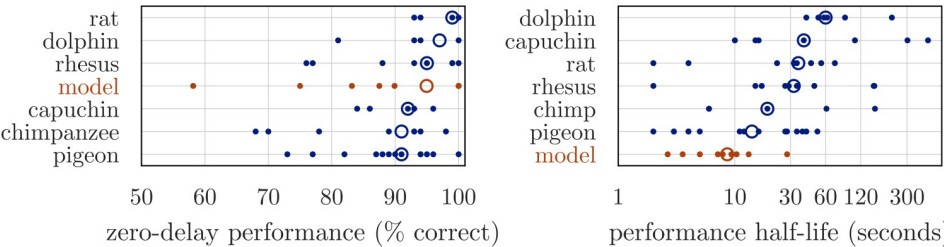

**Fig 14. Estimated zero-delay performance (left) and performance half-life (right) in DMTST across species.**
Performance half-life is defined as the delay for which performance drops from its zero-delay value to a value halfway
toward chance performance. Open circles are species medians. Empirical data are taken from [30]. We observe
significant differences in the rate of forgetting (performance half-life) between individual networks, a trend that we
also observe between individual animals (or experiments) within the empirical data. While the median mnemonic
performance of our networks is lower than the median performance of most species in [30], our high-performing
networks still outperform a significant number of individual monkeys, rodents, and birds, suggesting that our
cognitive networks operate in a biologically plausible WM regime.

Our best-fit parameters are consistent with the forgetting parameters reported in a recent
meta-analysis of animal mnemonic performance in the delayed match-to-sample (DMTST)
task [30], a DRT that assesses numerous aspects of working memory capacity. The resulting
dataset includes behavioral data from over 90 experiments, 25 species, and multiple delay
intervals. For each species in the dataset, the authors used an exponential curve to quantify the
relationship between DMTST performance and delay intervals. Unsurprisingly, the fitted
parameters varied significantly both across species and across experiments with the same spe-
cies, as shown in Fig 14. The baseline performance $B$ (characterized as "zero-delay perfor-
mance") was consistently high for all species, with median estimates varying between 58%
(chickadees) and 99.5% correct (rats) with a grand median of 93% correct. The forgetting time
constant $\tau_{half}$ (characterized as a "performance half-life", or the delay for which performance
has fallen halfway between zero-delay performance and chance performance) differed signifi-
cantly across species, with median estimates varing between 2.4s (bees) and 71s (dogs), with a
grand median of 27s. As shown in Fig 14, the performance of our cognitive network easily falls
within these ranges, with our median baseline performance $B$ = 95% resembling the grand
median baseline of 93%, and our median forgetting time constant $\tau_{half}$ = 8.6s most closely
resembling the forgetting rate of pigeons ($\tau_{half} \simeq 10s$). These correspondences speak to the
cognitive plausibility of our biologically-detailed model, showing that it produces behavior-
ally-plausible results despite the numerous low-level constraints we enforce in the network.

## 5 Discussion

Our goal in this paper has been to develop the "oracle supervised Neural Engineering Frame-
work" (osNEF), a method for training biologically-detailed spiking neural networks to realize
various dynamical systems that are relevant to cognition. We began by defining the relation-
ship between spike space, the dynamical pattern of action potentials generated by a popula-
tions of neurons, and state space, a vector-valued representational scheme for characterizing
neural dynamics. Building off the NEF, we decomposed synaptic weights into encoders and
decoders, and found parameters for these quantities such that neurons in a population exhib-
ited diverse tuning curves, effectively constructing a function basis for the desired dynamical
systems. We then presented a novel online learning rule for training encoders and decoders to
realize these tuning curves in the context of a larger spiking neural network. We also presented
an offline optimization procedure for training the time constants of synaptic connections that

helped account for nonlinear dynamics within the cell when realizing network-level dynamics. After presenting four neuron models that ranged from the simple and computationally-inexpensive to the complex and biologically-detailed, we showed that osNEF could be used to train neural networks that implement several cognitively-relevant dynamical systems. Specifically, our networks were populated by LIF neurons, Izhikevich neurons, Wilson neurons, or 4-compartment, 6-ion-channel Pyramidal cell reconstructions, and we trained them to compute the identity function, to multiply two scalars, to exhibit simple harmonic oscillation in two dimensions, and to save and load information with a working memory. Finally, we applied these methods to build a simple cognitive system that performs a DRT using biologically-detailed components, including Pyramidal cells, inhibitory interneurons, conductance-based AMPA and GABA synapses, and voltage-gated NMDA synapses. We tested our network's mnemonic performance by measuring the number of correct responses it returns as a function of delay length, then showed that this performance is comparable to animal performance in the DMTST [30]. In this section, we discuss our methods and results in terms of biological plausiblity, cognitive capacity, and usability, compare our methods to similar approaches, and present avenues for future research.

## 5.1 Biological plausibility

The central motivation of this paper was to introduce more biological realism into NEF-style networks and maintain their cognitive capabilities, allowing future researchers to investigate questions that relate low-level biological details to high-level cognition. Because osNEF operates exclusively in the spike space of neural activity and the state space of dynamical systems, our methods are agnostic about the internal structure of neuron and synapse models. This allows modellers to apply osNEF to a wide variety of neuron models, ranging from point neurons with a single state variable to multi-compartment models with complex state dynamics, using features of the synapses to manage intracellular nonlinearities and achieve the desired network-level dynamics. To demonstrate the utility of these features, we simulated neuron models with electrophysiologically plausible internal dynamics, connected them with higher-order synapses (including some whose dynamics couple with intracellular dynamics directly), and showed that they can perform a variety of cognitive operations. As discussed in Sec 5.4, these biological and functional capabilities are a significant extension of existing methods in computational neuroscience.

Despite these biological extensions, our networks still depart from biological realism in several respects. While our neuron and synapse models themselves conform to biology, the connectivity within the network is less constrained. For instance, all neurons in our networks connect to post-synaptic cells with both excitatory and inhibitory synapses, but Dale's Principle suggests that biological neurons exclusively release one type of neurotransmitter [31] (although some neuroscientists have questioned this principle, see [32]). More importantly, the networks presented here do not attempt to reproduce the statistics of neural connectivity between populations, to diversify neuron morphology (beyond small variations in compartmental geometry) or cell type (beyond Pyramidal cells and inhibitory interneurons), or to match other network-level anatomical details. These are important biological features that have been the focus of other projects concerned with biologically-detailed anatomical reconstruction (notably the HBP with respect to cortical microcircuits [2]), and these features may affect a network's ability to compute specific functions or perform particular cognitive operations. Fortunately, osNEF do not prohibit the inclusion of such feature; future work should investigate whether imposing these connectivity constraints pose a problem for our methods.

Another questionably-realistic aspect of our method is the online learning rule for updating synaptic weights, summarized in Eqs 8 and 9. Many components of these equations involve only local information, such as presynaptic activity, postsynaptic activity, and a state space error signal, and so are plausibly-accessible by individual neurons for learning. Previous work has argued for the biological plausibility of a learning rule based on these quantities [13]. However, one critical component in osNEF is the "oracle", a population of neurons built using standard NEF tools that generates the desired activities for our neurons. While it is possible that the brain contains "teacher" populations that supervise the weight updates within "student" populations, we are not aware of any empirical evidence that directly support the existence of such populations or validate Eq 8. We must therefore treat the components of osNEF that involve spike space supervision as biologically unfounded, and thus as a theoretical tool for constructing networks instead of a biological hypothesis.

Finally, the osNEF procedure for optimizing synaptic time constants is not intended to mimic a biological mechanism, but simply to find synapses that realized particular network-level dynamics given complex neuron-level dynamics. It is possible that future neuroscience research will reveal some optimization process in the brain that effectively selects which neurotransmitters, receptors, or network structure are used in a given system for the purpose of controlling temporal responses. Studies of highly-structured neural circuits, such as the granule-Golgi cells in the cerebellum [33], indicate that plasticity may interact with synaptic and intrinsic cellular responses (such as rebound firing [34] and inhibitory inputs [35]) to alter the temporal properties of the network. While the osNEF optimization via Hyperopt certainly would not reproduce the mechanisms of such an optimization, it might reproduce the results, with respect to the balance of time constants observed in the final network. This hypothesis is supported by S1 Appendix, which shows that the time constants discovered by osNEF often fall within biologically plausible ranges.

## 5.2 Cognitive capacity

To create cognitive models out of biologically-detailed neural networks, we trained our networks to implement linear dynamical systems described by control theory (Eq 6). Previous work with the NEF and SPA has shown that such an approach can be used to build extremely large brain models capturing a wide class of cognitive operations and reproducing behavioral data [3], justifying it as an appropriate framework for cognitive models. In this paper, we first showed that osNEF could produce neurons whose tuning curves were an effective function basis despite neural adaptation, spike noise, and the like. To further demonstrate this, we trained networks to perform a variety of operations common to cognitive systems, then combined several such operations together into a larger cognitive network. This network successfully performed a DRT, exhibiting an exponential forgetting curve that closely resembles the forgetting curves exhibited by simple animals (pigeons) performing a DMTST [30]. These mnemonic capacities are impressive given that (a) our network uses only hundreds of neurons in the memory populations, which is likely orders of magnitude fewer neurons than are present in the corresponding populations in behaving animals, and (b) our DRT task is significantly harder than the DMTST task used in the meta-analysis: our task included eight possible cue locations where the DMTST included only two; and our network used a cleanup memory to transform the remembered cue location into a behavioral output, where the DMTST presented the candidate cues again during the "choice" portion of the task, effectively "externalizing" the cleanup process. In conjunction with the broader successes of the NEF and SPA, we expect that osNEF can be used to train more sophisticated cognitive networks built from biologically-detailed components.

However, osNEF tools are limited in several respects. Networks trained using the standard NEF methods typically have less error than the results we reported in Sec 4.2. This is due, in large part, to the lack of biological constrains imposed by the default parameters implementing NEF networks in Nengo: such networks use point neurons and lowpass current-based synapses, leading to fewer cellular nonlinearities, and have firing rates approximately ten times higher than our simulations. Normal NEF methods also realize tuning curve distributions that provide a superior function basis. In standard NEF networks, neurons are biased by directly injecting a current into the "soma", allowing for precise control over the conditions under which a neuron will begin firing. Before the network is simulated, Nengo optimizes these bias currents and the encoding vectors (Eqs 1 and 2) such that neurons exhibit a wide range of x- and y-intercepts. osNEF does not use current injection to bias neurons: instead, it trains synaptic weights to achieve the target tuning curve distribution, making postsynaptic activities more dependent on noisy presynaptic spikes. In recurrent networks where feedforward input was selectively removed (the oscillator and the gated difference memory), we needed to introduce and train a dedicated *bias* population to stabilize recurrent activity. Finally, not all target tuning curves can be realized by a given neuron type: osNEF will fail if the targets have physiologically-implausible response curves or if a detailed neuron is tightly constrained by its morphology. Because ours is a black-box approach, the only way to know whether a neuron may be trained to reproduce a target tuning curve is through trial and error.

It is also worth noting that the dynamics realized in osNEF-trained neural networks are not exactly equal to the target dynamics: emergent neural dynamics depend upon the training signals used, the tuning curve distributions, the saturation of firing rates in biologically detailed neurons, and more, meaning that our networks only approximate the target systems. Sometimes, these constrains facilitated training, as when neural saturation (and resting-state activity) helped prevent explosion (decay) of amplitudes in the simple harmonic oscillator over long timescales (see Fig 9), a danger when using Eq 13 to implement an oscillator. Other times, these constraints complicated training, as when the noise introduced by spiking activity lead to imperfect representations of inputs or stored values in the gated working memory (see Figs 11 and 13), causing the memory to store incorrect values and to "forget" these values over time.

## 5.3 Usability

In developing osNEF, we strove to make our methods both broadly applicable and easy to learn, with the hopes that other researchers will apply them in their own projects. We used the Nengo ecosystem [9] because it is a scalable and flexible neuron simulator that has been used to build and train numerous neural and cognitive models, including some of the world's largest functional brain models [20]. To support a wide range of biological mechanisms, users may simulate neural and synaptic dynamics in either Python or NEURON. Python is appropriate for simulating simple models: this language has intuitive syntax and can be used alongside NumPy for efficient matrix operations [36]. NEURON is appropriate for simulating complex neuron models: this language is designed to efficiently simulate detailed cellular and synaptic mechanisms. Libraries of neural reconstructions written in NEURON are widely available from online repositories like the Allen Cell Types Database [37] and the Neocortical Microcircuit Collaboration Portal [38], and our interface allows researchers to connect these models into the Nengo ecosystem for functional modelling, as we did with the Durstewitz model [19]. That said, training neural networks with osNEF requires significantly more effort and simulation time than standard Nengo networks. Users must choose appropriate training signals, build a parallel "oracle" network to generate the target activities, and simulate the network

over time to engage online learning. The challenges posed by different neuron types and target dynamics makes automating this process difficult, so the user must complete these steps manually. However, we believe that the broad applicability of osNEF justify these challenges for the motivated researcher.

## 5.4 Comparison to other methods

The Neural Engineering Framework is the theoretical core of osNEF: from it, we borrow (a) the distinction between spike space and state space, (b) the decomposition of weights into encoders and decoders, and (c) the use of control theory to specify target dynamics. We also use a number of standard NEF tools, including (d) the PES learning rule, and the specification of target networks via (e) least-squares optimization of decoders and (f) distribution of encoders, gains, and biases. osNEF extends the NEF by (1) redefining encoders as a tensor over pre-synaptic neurons, postsynaptic neurons, and state space dimensions, (2) introducing an online learning rule to update encoders and decoders based on state space error, spike space supervision, and Hebbian activity, and (3) optimizing the time constants of synapses to realize network-level dynamics while accounting for adaptive neuron-level dynamics. Many of these techniques bear a resemblance to other research in computational neuroscience, both in motivation and in practice. A central theme of modeling paradigms ranging from FORCE learning [8] to efficient balanced networks (EBN, [39]) is to describe cognitive algorithms in terms of the dynamics of a latent state variable $\mathbf{x}(t)$ represented by neural activity, then train neural connection weights such that the network behaves like a target dynamical system. However, the techniques by which networks are constructed and trained varies significantly between these paradigms: osNEF borrowed several of these techniques in extending the NEF.

In the full-FORCE method [40], the recurrent activities of a neural network are trained with the aid of a parallel target-generating network that is driven by the desired output of the system. full-FORCE target networks have random internal connectivity: when driven by the target dynamics, such networks produce activities that include both a chaotic component (from the random recurrent connectivity) and a desired component (from the driving input). Such activities, the authors hypothesize, is a suitable basis for realizing non-trivial dynamics, especially when combined with an optimized readout filter. In their paper, the authors show that a recursive least squares optimization process, which compares the target activities with the activities of the task-performing network, may be used to train recurrent weights in the later and reproduce a wide variety of dynamics. Our "oracle" populations are also driven to exhibit the target dynamics and used as a resource when learning recurrent weights. However, our oracle does not rely on random connectivity: we use the NEF to specify weights that guarantee that "tar" will exhibit the target dynamics as well as the spike variability that promotes a robust function basis for computation (through the principled distribution of encoders and tuning curves). By specifying the oracle in this way, our approach greatly simplifies the relationship between the task-performing and target networks: they are both driven by the same external inputs, rely solely on recurrent activity to generate the target dynamics, and should exhibit the same activities neuron-by-neuron. This eliminates a great deal of parameter tuning required by full-FORCE, and provides a clear conceptual pictures of how the target network supervises the task network. Furthermore, osNEF uses a local, online, error driven learning rule, while the RLS approach used in full-FORCE is a global, iterative update that is very unlikely to be implemented by brains.

A recent extension of EBNs to nonlinear adaptive control theory [41] also bears many similarities to osNEF. In this paper, the authors realize nonlinear dynamics in a recurrently-connected population of spiking LIF neurons using a state space teacher and an online learning

rule. Many mathematical similarities exist between this approach and the NEF, especially with regards to the PES rule (Eq 9) that we use for error-driven learning in the state space. The authors convincingly demonstrate their ability to learn nonlinear dynamics, including a bistable attractor and rhymic walking motions derived from motion capture data, and their networks are similarly constrained to low firing rates, small numbers of neurons, and irregular spiking. However, where their work (and EBNs in general) focuses on using fast inhibitory connections to create an efficient, balanced coding scheme, our work focuses on implementing all of the required components in biological detail. While the authors describe how biological components could be used to implement their methods (AMPA/GABA-A synapses for fast connections, NMDA/GABA-B synapses for slow connections, and nonlinear dendrites for the basis functions), dealing with the nonlinear, non-instantaneous dynamics imposed by compartmental neurons and conductance-based synapses often requires significant theoretical extensions, even when the underlying framework can already model nonlinear dynamical systems.

Recent papers using both FORCE [42] and EBN [43] have implemented linear (and sometimes nonlinear) dynamics in neural networks populated with biologically detailed neurons. This raises the question: if simulation of functional networks with complex neurons is already possible with other techniques, why use osNEF? We believe there are several features that make osNEF a novel, worthwhile contribution to this field. Broadly speaking, there are important theoretical differences between the underlying frameworks, FORCE, EBN, and NEF. Even if we assume that all three methods solve similar problems with similar performance, there is significant value in developing and presenting an NEF-based method that is comparable to FORCE- and EBN-based methods. The NEF modeling community is quite large, so validating a method for training biologically-detailed functional models within this framework is important, independent of similar successes with FORCE and EBN. Furthermore, given the complexity of training biologically-detailed networks, we believe that the research community will benefit from the existence of multiple methods that tackle this problem in different ways. Future work should compare these methods, identify their relative strengths and weaknesses, and develop new methods that build upon their successes.

There are also significant differences between these three methods with respect to biological plausibility and computational capacity. In the FORCE paper [42], the authors simulate spiking LIF, Theta, and Izhikevich neurons, showing that they can produce a number of dynamical systems, including oscillators, chaotic systems, songbird calls, and episodic memories. These methods use a variety of current-based synapse models (exponential, double exponential, and alpha) and respect Dale's principle. In the EBN paper [43], the authors simulate point neurons that include Hodgkin-Huxley-type ionic currents, showing that they can produce several types of one-dimensional integrators that statistically reproduce empirical patterns of neural activity. Their methods also use double-exponential synapses, but do not respect Dale's principle. In our paper, we simulate three point neurons and one pyramidal cell reconstruction with four compartments and six ionic currents. We show that our trained networks reproduce four linear dynamical systems (feedforward and recurrent networks) and perform a cognitive task at performance levels comparable to simple animals. Our methods use both current-based double-exponential synapses and conductance-based, voltage-gated NMDA synapses; we do not respect Dale's principle. We feel confident in claiming that we have (a) simulated neurons with significantly more biological realism than the FORCE and EBN papers, except with respect to Dale's principle (but see below), and (b) demonstrated that our trained networks perform a wider variety of computations than [43] and a comparable number to [42] (excepting nonlinear systems).

Other researchers have extended the NEF with the goal of increasing the framework's biological plausibility. Stöckel has developed numerous methods for training neural networks to realize computationally-useful dynamical systems. One such method simulates networks of multi-compartment LIF neurons in which synaptic connections are decomposed into excitatory and inhibitory components, each characterized by an appropriate equilibrium potential. By selectively placing these synapses on different compartments and optimizing excitatory and inhibitory weights, Stöckel shows that these dendrites can effectively compute functions like four-quadrant multiplication [44]. Other work applies this E-I optimizer to a functional model of eyeblink conditioning in the cerebellum's granule–Golgi microcircuit, demonstrating that anatomically-detailed spatial connectivity can be profitably incorporated into NEF models while respecting Dale's principle [45]. These methods nicely complement the osNEF: where Stöckel focuses on adding biological details to the connections between groups of neurons (managing E-I balance, targeting dendrites, and reproducing spatial structure), we focus on adding biological details to the internal dynamics of the underlying components (the neuron model and the synapse model). Future work that combines these techniques would greatly enhance the biological plausibility of NEF networks, which have previously been criticized for lacking particular biological features.

The use of high-order synapses to control network-level dynamics has also been explored using the NEF and EBN. Voelker has derived a method for computing the parameters of high-order synapses that, when simulated in networks of simple neurons, preserves state space dynamics in the network. These NEF-style techniques can be used to construct models that encode rolling windows of input history, and the resulting neural activities closely resemble the mnemonic responses of time cells in the cortex [12]. In contrast to osNEF, these analytical techniques guarantee a solution, but rely on certain assumptions that are violated once sufficient biological detail is included: specifically, if the dynamics internal to the neuron model dominate the synaptic dynamics, or if the synaptic dynamics are coupled to the cellular dynamics. EBN has also utilized synaptic dynamics to account for the nonlinear dynamics of complex neurons, but relies on a significantly different optimization processes for discovering those synaptic dynamics. In [43], the authors use a form of system identification that (a) drives the neuron model with a specific random process, (b) analyses the resulting voltage traces when the neuron spikes, (c) calculates the average action potential waveform, (d) takes its temporal derivative, and (e) convolves it with an exponential function. In our networks, where cellular adaptivity is a major force, where voltage-gated synapses depend on intracellular activity, and where the dynamics of distinct geometric compartments are coupled, applying these types of analytical techniques become difficult or impossible. Still, these methods demonstrate the utility of optimizing synaptic time constants for network-level dynamics.

## 5.5 Future work

This paper has introduced several methods for incorporating biological detail into cognitive neural networks. While we have demonstrated the functional capacity of osNEF and applied it to train network that perform simple cognitive tasks, the scientific utility of developing such techniques lies in an enhanced ability to investigate the relationship between low-level biological features and high-level cognition. Here, our goal was to develop osNEF and show that it could be used to build simple cognitive models; future work should use these models to form and test specific hypotheses related to pharmacology, neurological disorders, and other phenomenon where biology directly affects cognitive abilities. For example, the neurotransmitter Dopamine (DA) plays an important role in maintaining stable delay-period activity in working memory tasks, but there seems to be a "Goldilocks zone" for DA levels: having either too much

or too little neurotransmitter impairs cognitive performance [46]. The pyramidal cells in our cognitive neural network were trained and tested at a constant level of DA; by varying these levels, we could attempt to reproduce this phenomenon or better characterize its dynamical origins. What's more, we could use our model to help design pharmacological interventions for individuals who live with chronically-low levels of DA.

## 6 Conclusion

Functional capacity and biological plausibility are two important criteria for neural network models that seek to clarify the relationship between low-level mechanisms and high-level behavior. In this paper, we developed osNEF, a method to train biologically-detailed spiking neural networks to realize cognitively-relevant transformations and dynamics. osNEF utilizes an online learning rule that (1) combines insights from several theoretical frameworks, (2) includes error-driven, supervised, and Hebbian components, and (3) can be applied to a wide variety of neuron and synapse models. We demonstrated the utility of osNEF by (a) showing that the neural activities of a trained network form an appropriate function basis for dynamic computation, (b) building several functional networks that perform cognitively-useful operations with high accuracy, and (c) combining these operations into a larger cognitive network that performs a simple working memory task. This cognitive network is built from numerous biologically-detailed components, including Pyramidal cells, inhibitory interneurons, conductance-based GABA synapses, and voltage-gated NMDA synapses, and performs a delayed response task with mnemonic performance comparable to simple animals such as pigeons. We concluded by discussing the biological realism of osNEF, assessing its cognitive capacity and usability, and comparing it to similar methods in the literature. Future work should focus on applying osNEF to other cognitive systems in the brain, with a particular focus on studying how low-level biological features (or deficits thereof) contribute to high-level cognitive capacity.

## Supporting information

**S1 Appendix. Synaptic time constants.** Includes a table that reports the time constants used for each of the networks in Sec 4, as well as a figure that reports the distribution of time constants obtained when repeatedly running Hyperopt.
(PDF)

**S2 Appendix. Performance versus noise.** Includes an additional experiment that investigates how the performance of osNEF-trained networks degrades as external noise is introduced into the system.
(PDF)

## Author Contributions

**Conceptualization:** Peter Duggins, Chris Eliasmith.

**Formal analysis:** Peter Duggins, Chris Eliasmith.

**Investigation:** Peter Duggins.

**Methodology:** Peter Duggins, Chris Eliasmith.

**Project administration:** Chris Eliasmith.

**Software:** Peter Duggins.

**Supervision:** Chris Eliasmith.

**Validation:** Peter Duggins, Chris Eliasmith.

**Visualization:** Peter Duggins.

**Writing – original draft:** Peter Duggins.

**Writing – review & editing:** Chris Eliasmith.

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
