## [Decision Letter · Decision Letter 0]

24 May 2022

Dear Mr Duggins,

Thank you very much for submitting your manuscript "Constructing Functional Models from Biophysically-Detailed Neurons" for consideration at PLOS Computational Biology.

As with all papers reviewed by the journal, your manuscript was reviewed by members of the editorial board and by several independent reviewers. In light of the reviews (below this email), we would like to invite the resubmission of a significantly-revised version that takes into account the reviewers' comments.

We cannot make any decision about publication until we have seen the revised manuscript and your response to the reviewers' comments. Your revised manuscript is also likely to be sent to reviewers for further evaluation.

Sincerely,

Michele Migliore

Associate Editor

PLOS Computational Biology

Samuel Gershman

Deputy Editor

PLOS Computational Biology

Reviewer's Responses to Questions

**Comments to the Authors:**

Reviewer #1: This work by Duggins and Eliasmith introduces an (unnamed) biologically plausible learning rule to train biophysically detailed spiking neural networks to display the dynamics of various linear dynamical system, along with performing a Delayed Matching to Sample Task. The networks were all successfully trained, and the paper is well written. However, I’m unsure of the robustness of the results, the significance, and the novelty. Further, some of the choices in network/algorithm design were a bit perplexing, and I had trouble finding Figure 3 for the attached paper! I outline my points below:

Major Points

1. It’s somewhat frustrating that there’s no name for the method to train biophysically detailed neuron models. I understand that this technique borrows from many sources, such as the NEF, but it’s hard to keep focus when it’s just referred to in third person and never by any proper name like “Time NEF” or “encoder-training boosted NEF”.

2. I think this paper would benefit substantially from a numerical demonstration of a non-linear dynamical system. There are techniques (such as the efficient coding based spiking networks) for which non-linear dynamics are difficult. Some justification/testing is required for the statement “the method remains the same for nonlinear system”. In particular, I have some doubts that the non-linear dynamics of something like the Lorenz or Rosler systems will be reproduced when the neural tuning curves are approximated by ReLus, but I could be wrong.

3. The choice of the ReLU. Why use a ReLu vs a type-I tuning curve? These are the tuning curves that would be expected for type-I neurons on a SNIC bifurcation. Also, the LIF neuron model here seems a weird choice, since there’s already a derivable tuning curve.

4. Because the synaptic time constants were part of the optimization, I would like to see a table that shows their values for all tasks considered. This partially shown in the representation task but we need a table for every distinct network trained/task, since the time constants always differ.

5. Stability of the Hyperopt scheme: If you run the hyperopt scheme for the same network/task pairing multiple times, will you arrive at the same synaptic time constants? Is the performance in anyway monotonic for the decay time with a fixed rise time?

6. One of the core problems I have with the procedure to optimize the time constants. Many networks are engaged in more than a single task, and don’t optimize the time constants at all, or at least through no known mechanism. In fact, there’s typically a narrow range for specific synapse types. I’m not sure of the biological plausibility of this approach.

7. Figure 3 was not labeled in the attached PDF! I’m not sure if this was down to the authors, or the submission system. The figures are also out of order at the end. I think what's happened is the figures were uploaded possibly out of order or were not explicitly labeled in the submission system with the right figure index.

8. Equation 12, while is an oscillator, is of a very special type. It is non-hyperbolic, meaning all amplitudes for the oscillator are stable. So if simulated long enough, the amplitude should decay, or explode. I’m a bit surprised there’s stability over 100s, I would show the amplitude of the cycles at the end of the 100s.

9. I’m not sure what the big picture here is in regards to the biophysical detail. It seems like it’s largely pointless in terms of any particular neuron being better or worse at specific tasks, as no one model really outperforms any other.

10. Reservoir methods like FORCE/echo state training can also use biophysically detailed models, since the learning algorithms are agnostic towards the node dynamics used. Further, there is work performed by the Deneve group that also extends the EBN approach to Hodgkin-huxley type models with linear dynamics. So, to what extent is the training of biophysically detailed neuron models even novel anymore? This seems to be possible now with a variety of techniques.

11. As far as I can tell, there’s no noise included in the networks here at all. Would any of the network dynamics still be trainable if the neurons each received a white noise process?

Minor Points

1. I would agree that some cellular features (line 46) are not useful for information processing, I just would not explicitly include the examples of mitosis/energy transport the authors make. Mitosis for example is necessary in neurogenesis in the dentate gyrus. To what extent its regulated by information processing requirements remains to be investigated. I would just be careful here and remove the examples.

2. Is equation (3) the formal cross product between two vectors, or just the matrix product? I would not use the X symbol when mentioning two vectors. The encoders introduced as a tensor later on, so I would just make it explicit what this operation is.

3. delayed response task -> Delayed Response Task (DRT)

Reviewer #2: Referee report on: Constructing Functional Models from Biophysically-Detailed Neurons (PCOMPBIOL-D-22-00598)

Submitted to: PLOS Computational Biology

In this paper, the authors developed a method to train biologically-detailed spiking neural networks with the ability to realize different cognitively-relevant dynamical systems.

The proposed method was tested in four distinct neuron models: the LIF neuron, the Izhikevich neuron, the Wilson neuron, and a Layer V Pyramidal Cell reconstruction. The results of the tests include populations of neurons produced with the desired properties and representative capacities, synaptic connections trained to compute specific functions or dynamics in networks of complex neurons, and a biologically-detailed model that performs a working memory task successfully trained by the method developed in this work.

The authors used inspiration from distinct fields to create this work, namely from Neural Engineering Framework, supervised learning, and online Hebbian weight updates. The resulting training method can be applied to a wide variety of neuron and synapse models, and potentially to other cognitive systems in the brain.

In my opinion, this manuscript presents an innovative framework to train biologically-detailed spiking neural networks and, eventually, more general brain models. The manuscript is clearly written, well organized, and it has adequate discussions. The references are also adequate and up to date. Hence, I recommend acceptance for publication.

Reviewer #3: Constructing Functional Models from Biophysically-Detailed Neurons — Major revision

The authors present a method for building functional neural networks that can perform foundational computations that can be used to generate models of cognitive tasks, and doing so with biologically detailed neuron models. The authors’ framework includes a biologically motivated online learning rule, an offline learning rule to tune the synaptic dynamics, and the ability to incorporate neuron models of varying biological detail. The authors approach builds on the well-established neural engineering framework (NEF), and can thus exploit the known strengths of that approach. The authors illustrate the successes of their approach on a number of benchmark tasks/computations.

Many aspects of this work are commendable. Broadly, the authors’ focus — developing methodology for bringing additional biological detail into networks for modeling cognition — is an important and under studied one, making this work important and appropriate for this journal. Additionally, the authors’ approach of tuning the synaptic dynamics (although biologically implausible) is an under studied consideration in constructing neural networks. It sounds as though the authors have devoted considerable attention towards creating an easy-to-use computational framework that easily connects existing frameworks (e.g. NEURON and Nengo), and for that reason, this work will be a benefit to the community, allowing researchers to explore and create network models with differing levels of biological detail. Finally, the results appear mostly sound and are valuable — understanding the relative performance of neuron models of varying complexity on a battery of tasks is an important scientific advance.

However, the manuscript in its current form needs significant work, and thus at this time, has received a response of ‘major revision’. The primary problem is the clarity — the manuscript is extremely difficult, and sometimes impossible, to follow.

Clarity issues came in two flavors: 1. General lack of clarity, e.g. insufficiently explaining details of the model or the results; 2. Lack of clarity regarding which elements of the model do or do not correspond to existing conventions from the NEF. Regarding the second point, both the potential of the authors’ approach as well as the challenges of clarity present in the manuscript stem from the approach’s closeness to the NEF; this allows the authors the ability to leverage the successes of the NEF but also make it unclear which parts of the framework are not standard NEF choices (i.e. which components are novel and/or which components have deviated from the NEF for specific reasons) and also which parts ARE standard NEF choices, and are therefore being used to exploit the known power of the NEF.

I provide a few examples of each type of clarity issue, to give the authors a sense of my concern. This list is not meant to be exhaustive. The burden of identifying all of these issues should not rest on a reviewer, but this list merely seeks to serve as signposts to inform a resubmission, which I will be happy to serve as a reviewer for, because as I wrote, the results themselves are very interesting.

General lack of clarity

- Network schematic figures (e.g. Figs 2,5,8,10,12) are very difficult to follow, and lack critical descriptions to understand them. For example, what is H in each? (I am left to assume this refers the standard engineering usage of H as an impulse response function, but this should be made explicit.) Figure 12 is particularly mysterious. Broadly, all of these figures are a bit on the basic side, and the model design principles are quite complex. More detailed figures would help to clarify the details of the design procedure. Perhaps adding some plots of the different signals at each stage (e.g. x, each neural population, etc.) would help make the model more clear.

- Another figure clarity example: In Fig. 4 top and middle, what are the different lines? Different model neuron classes or different neurons in a single network? It’s very difficult to parse those lines. If its different model neurons, there are 4 example model types (by my count) but it looks like there are more than 4 lines?

- Related: multiple figure captions are fairly terse. For example, the Figure 7 caption doesn’t seem to be related to the actual content of Figure 7 in any way that was clear to me.

- Several terms are introduced without explanation. What is hyperopt? (Of course I can google this myself, but shouldn’t have to). What is the PES learning rule (i.e. what does PES stand for? Never defined).

- The basics of the most simple model structure are difficult to follow (Section 3.2) and this is not helped by Figure 2 or the mathematical notation. a_pre is clearly designated as a presynaptic population, but what is a_pop? Is this population of neurons recurrently connected? In addition to an improved Figure 2 making the model architecture more clear, a more thorough treatment of the model details would make it more clear. Relatedly, the model equations are a bit scattered all over the place, because of the way the model is built up. It might be appropriate at 3.2 to describe the model in full detail (even revisiting things that have already been discussed) so all of the model components can be understood in relationship to each other.

- The authors seem to use the sections ‘Representation’, ‘Computation’, and ‘Application’ as anchors, but to me, this did not incrementally introduce the success of their approach, as I think the authors had intended. For example, the distinction between representation and computation was not clear to me, and the clarify was not aided by the figures. So either a restructuring of the results sections should happen or the existing structure should be explained more clearly.

Relationship to the NEF

- The online decoder learning rule is different than the NEF decoder learning rule (I think). Is this correct? If so, this should be more explicit, and also the decoder learning rule from the NEF should be stated explicitly so this distinction can be made more clear.

- It’s difficult to understand how the NEF is used (apart from inspiring the encoder/decoder convention) within the authors’ framework. Specifically, it seems that the NEF is used as a target for the encoders of the ‘learner’ network. How exactly this all works together (how is the target network constructed and how is it used for the learning network?) is difficult to follow from the manuscript (and also not helped by the figures). (Note that the comparison the authors provided in the Discussion between their approach and the full-FORCE approach of DePasquale et al, e.g. as a target generating and learner network, was helpful in bringing this key network design aspect into the light, but still, it would help to make this much more clear in any event, and especially much earlier in the manuscript than the Discussion, probably in Section 3.2).

- One element that is a bit unique about the NEF framework, and that the authors’ framework seems to share, is that the models are often described as having the state value x input into the system and the computations are done as operations on this representation. To me, this is often a somewhat foreign perspective and differs from the more typical perspective of neural networks these days where a state variable is contained implicitly within a neural population that potentially receives other external inputs. To me, clarity could be improved by walking a bit more slowly through the model architecture and to specifically drawn attention to and clarify this novel aspect of NEF-like frameworks.

**Have the authors made all data and (if applicable) computational code underlying the findings in their manuscript fully available?**

Reviewer #1: Yes

Reviewer #2: None

Reviewer #3: Yes

PLOS authors have the option to publish the peer review history of their article (what does this mean?). If published, this will include your full peer review and any attached files.

Reviewer #1: No

Reviewer #2: No

Reviewer #3: No
---

## [Decision Letter · Decision Letter 1]

30 Jul 2022

Dear Duggins,

We are pleased to inform you that your manuscript 'Constructing Functional Models from Biophysically-Detailed Neurons' has been provisionally accepted for publication in PLOS Computational Biology.

Best regards,

Michele Migliore

Associate Editor

PLOS Computational Biology

Samuel Gershman

Deputy Editor

PLOS Computational Biology

Reviewer's Responses to Questions

**Comments to the Authors:**

Reviewer #1: Overall, the authors have done a good job in addressing my previous criticisms. I only have minor comments that they're free to change or not. I recommend accepting the manuscript for publication.

1) I would keep the description of variables immediately after equation (7), rather than after the joint equation (9). A reader would see equation (7) first and would wonder what the terms are.

2) I got a bit lost with all the acronyms. A table might help?

3) In equation (12), I would add a quick description that u_1 and u_2 are the two scalar components of the vector u

4) Just as a side note, I would be curious to see the RMSE in Figure 15 correlated with the resulting time-constants for the different tasks.

5) I couldn't find the github link in the manuscript, but then I realized it's embedded in with a link. Maybe have the full URL. It'll just be a bit easier for readers to find.

Reviewer #2: Referee report on the revised version of "Constructing Functional Models from Biophysically-Detailed Neurons" (PCOMPBIOL-D-22-00598-R1)

Submitted to: PLOS Computational Biology

In this revised version of the paper "Constructing Functional Models from Biophysically-Detailed Neurons", the authors introduced several changes and explanations that clarify and made their work easier to understand, especially with regard to figures and network diagrams. The referees’ comments were all analysed and, in my opinion, the quality of the paper has improved a lot.

As I considered in my first revision, this paper presents an innovative framework to train biologically-detailed spiking neural networks, and so it is quite interesting. It is well organized and has good discussions. Therefore, I continue to recommend acceptance.

Reviewer #3: The revised manuscript has addressed all of my concerns from the initial submission. The clarity has been greatly improved by revisions to Figure 2 and Section 3 (especially Section 3.2). The revision does a better job clarifying what is new to the authors' approach and what are standard to NEF.

**Have the authors made all data and (if applicable) computational code underlying the findings in their manuscript fully available?**

Reviewer #1: Yes

Reviewer #2: None

Reviewer #3: Yes

PLOS authors have the option to publish the peer review history of their article (what does this mean?). If published, this will include your full peer review and any attached files.

Reviewer #1: No

Reviewer #2: No

Reviewer #3: No

---

## [Editor Report · Acceptance letter]

23 Aug 2022

PCOMPBIOL-D-22-00598R1 

Constructing Functional Models from Biophysically-Detailed Neurons

Dear Dr Duggins,

I am pleased to inform you that your manuscript has been formally accepted for publication in PLOS Computational Biology. Your manuscript is now with our production department and you will be notified of the publication date in due course.

With kind regards,

Zsofia Freund
